# Research on the Buckling Behavior of Functionally Graded Plates with Stiffeners Based on the Third-Order Shear Deformation Theory

**DOI:** 10.3390/ma12081262

**Published:** 2019-04-17

**Authors:** Hoang-Nam Nguyen, Tran Cong Tan, Doan Trac Luat, Van-Duc Phan, Do Van Thom, Phung Van Minh

**Affiliations:** 1Modeling Evolutionary Algorithms Simulation and Artificial Intelligence, Faculty of Electrical & Electronics Engineering, Ton Duc Thang University, Ho Chi Minh City 700000, Vietnam; nguyenhoangnam@tdtu.edu.vn; 2Training Department, Le Quy Don Technical University, Hanoi City 100000, Vietnam; ttan3105@gmail.com; 3Faculty of Mechanical Engineering, Le Quy Don Technical University, Hanoi City 100000, Vietnam; doanluat.vn@gmail.com (D.T.L.); minhpv.mta@gmail.com (P.V.M.); 4Center of Excellence for Automation and Precision Mechanical Engineering, Nguyen Tat Thanh University, Ho Chi Minh City 700000, Vietnam

**Keywords:** buckling, FGM, stiffener, FEM, third-order shear deformation theory

## Abstract

This paper presents a finite element formulation to study the mechanical buckling of stiffened functionally graded material (FGM) plates. The approach is based on a third-order shear deformation theory (TSDT) introduced by Guangyu Shi. The material properties of the plate were assumed to be varied in the thickness direction by a power law distribution, but the material of the stiffener was the same as that of the one of the bottom surface where the stiffener was placed. A parametric study was carried out to highlight the effect of material distribution, the thickness-to-width ratio, and stiffener parameters on the buckling characteristics of the stiffened FGM plates. Numerical results showed that the addition of stiffener to the FGM plate could significantly reduce the weight of the FGM plate but that both the FGM plates with and without stiffener had equally high strength in the same boundary condition and compression loading.

## 1. Introduction

The common way to achieve higher strength for functionally graded material (FGM) plates and shell without stiffeners is to either increase the thickness of this structure or to add stiffeners. The weight of the unstiffened structure will become higher with increasing thickness, but reinforcement with stiffener will reduce the weight as well as the cost of this structure. For this reason, using stiffeners is the best method in special cases such as ship building, bridge construction, aerospace, marine, and so on.

To give more useful information about the application in practice, the buckling behavior of composite structures has received much attention from scientists. Broekel and Gangadharaprusty [1] used experimental and theoretical solutions to study the mechanical responses of stiffened and unstiffened composite panels subjected to a uniform transverse loading. Liu and Wang [2] explored the elastic buckling of a plate reinforced by stiffener under in-plane loading. ANSYS modeling was applied to find the optimal height, number, and arrangement of stiffeners. Ueda et al. [3] proposed an analytical approach to research the buckling and deflection responses of a stiffened plate, which has a deflection under out-of-plane and in-plane loads. Danielson et al. [4] presented a combination of von Karman and nonlinear beam theories to predict the buckling behavior of a stiffened plate subjected to axial compression. Using the finite element method, Jiang and co-workers [5] found that the second-order 2D solid element gave accurate results for the buckling problem of unstiffened and stiffened plates. The ABAQUS solution was used by Hughes et al. [6] to examine the buckling behavior of T-stiffened panels subjected to uniaxial compression and lateral pressure. Pavlovcic et al. [7] investigated the buckling problem of imperfect stiffened panels using numerical simulation and tests. 

Nowadays, with the development of science and technology, many fields require machine details and structures working in harsh environments, such as high temperature, large abrasion, complicated loads, and so on. Therefore, in order to satisfy this demand, the material industry must develop and find new materials specialized for practice areas. Thus far, many new materials have been created to meet this requirement, such as the typical functionally graded material invented by Japanese scientists in 1984. Due to its outstanding advantage of being able to perform in a high-temperature environment as well as its large load capacity, anti-radiation, and anti-corrosion properties, this material has been widely applied in many important areas, such as nuclear science, medicine, chemistry, mining, and so on.

Javaheri and Eslami [8] explored the thermal buckling behavior of rectangular FGM plates on the basis of classical plate theory and closed-form solutions. This approach was also used by Shariat and Eslami [9] to study the thermal buckling of imperfect FG plates. The thermal postbuckling response of FGM skew plates based on the finite formulation was adopted by Prakash et al. [10]. Recently, Moita et al. [11] presented a formulation for buckling and nonlinear analysis of FGM plates subjected to mechanical and thermal loadings. Based on a semianalytical approach, Dung and Nam [12] analyzed the nonlinear dynamics of eccentrically stiffened functionally graded circular cylindrical thin shells under external pressure and surrounded by an elastic medium. In [13], Yu and his co-workers investigated the thermal buckling of functionally graded plates with internal defects in which the extended isogeometric analysis was fully exploited. For this type of problem, Dung and Nga [14] also discovered the thermomechanical postbuckling of eccentrically stiffened sandwich plates on elastic foundations subjected to in-plane compressive loads, thermal loads, and thermomechanical loads at the same time. The buckling of parallel eccentrically stiffened functionally graded annular spherical segments were studied through the Donnell shell theory and smeared stiffeners technique by Nam et al. [15]. Bohlooly and Fard [16] introduced new results for buckling and postbuckling of concentrically stiffened piezo-composite plates on elastic foundations. Chi and Chung [17] found an analytical solution based on the classical plate theory for the mechanical behavior of fully simply supported FGM plates subjected to transverse loading. Nam et al. [18] presented finite modeling for the mechanical buckling of cracked stiffened FGM plates based on the first-order shear theory of Mindlin. Trabelsi et al. [19] explored the thermal postbuckling behavior of functionally graded plates and cylindrical shells using four-node element based on a modified first-order shear deformation theory. These authors continued using this approach to investigate the thermal buckling of functionally graded plates and cylindrical shells [20]. Chen et al. [21] studied the buckling and bending behavior of a functionally graded porous plate, with the formulations based on the first-order shear theory and the Chebyshev–Ritz method.

There are many plate theories (from classical to higher-order shear deformation theories) that we can apply to analyze the mechanical behavior of structures made of anisotropic and isotropic materials. The Shi shear deformation theory [22] is a higher-order shear deformation theory that has many advantages, as discussed in detail in the literature [22,23], and it gives an accurate solution for the analysis of shear flexible plates. This theory can be developed to solve the nonlinear constitutive behavior of materials [24] and problems of materials with misfitting inclusions [25].

In all of the above published works, many results and conclusions were achieved on the buckling behavior of unstiffened and stiffened plates. However, a detailed study on the percentage weight loss of a stiffened FGM plate compared with a unstiffened one (these two plates having the same buckling strength) has not been done despite this research being very important in structural design and manufacturing. This paper presents a finite element formulation for mechanical buckling responses of stiffened FGM plates based on the G. Shi shear deformation theory. New numerical results were computed to examine the effect of different parameters on the buckling problem of stiffened FGM plates. This study demonstrates clearly the decrease in the weight of a stiffened FGM plate compared with a unstiffened one. Furthermore, this work studied the effects of the distance between two stiffeners on the buckling loads of a stiffened plate in order to determine if the distance results in a buckling strength higher or lower than that of a plate with one central stiffener.

This paper is structured as follows. Section 2 shows the finite element method using the Shi shear deformation theory for the buckling problem of stiffened FGM plates. The numerical results of the buckling loads for stiffened FGM plates are computed and discussed in Section 3. Section 4 gives the main conclusions of this study.

## 2. Finite Element Formulation for Mechanical Buckling of Stiffened FGM Plates

We considered a functionally graded material stiffened plate composed of ceramic and metal phases. The material on the top surface of this plate was full of ceramic and was graded to metal at the bottom surface of the plate by the power law distribution. The stiffener was placed at the bottom surface, and it was full of metal. This meant that the material of the stiffener and the bottom surface was the same. 

The thickness, length, and width of the plate have been noted as *h*, *a*, and *b*, respectively, while the depth and width of the stiffener are *h_s_* and *b_s_*, respectively, as sketched in Figure 1. The material was graded by the power law distribution, and it was used for describing the volume fraction of the ceramic (*V_c_*) and the metal (*V_m_*) as follows [23,26]:(1a)$$V_{m}+V_{c}=1$$
and
(1b)$$V_{c}(z)={\left(\frac{1}{2}+\frac{z}{h_{p}}\right)}^{n}$$
where, *h* is the thickness of the plate; *n* is the gradient index (*n* ≥ 0); *z* is the thickness coordinate variable (−*h*/2 ≤ z ≤ *h*/2); and subscripts *c* and *m* represent the ceramic and metal constituents, respectively.

In this study, the Young’s modulus *E* and the Poisson’s ratio ν change through the *z*-direction as follows [23,26]:(2)$$E\left(z\right)=E_{m}+\left(E_{c}-E_{m}\right)V_{c};\;\nu \left(z\right)=\nu_{m}+\left(\nu_{c}-\nu_{m}\right)V_{c}$$

Using the Shi shear deformation theory [22,23], the FGM plate model has the following displacement field (*u, v, w*):(3a)$$u\left(x,y,z\right)=u_{0}\left(x,y\right)+\frac{5}{4}\left(z-\frac{4}{3h^{2}}z^{3}\right)\phi_{x}\left(x,y\right)+\left(\frac{1}{4}z-\frac{5}{3h^{2}}z^{3}\right)w_{0,x}$$
(3b)$$v\left(x,y,z\right)=v_{0}\left(x,y\right)+\frac{5}{4}\left(z-\frac{4}{3h^{2}}z^{3}\right)\phi_{y}\left(x,y\right)+\left(\frac{1}{4}z-\frac{5}{3h^{2}}z^{3}\right)w_{0,y}$$
(3c)$$w\left(x,y,z\right)=w_{0}\left(x,y\right)$$
where *u*_0_, *v*_0_, and *w*_0_ represent the displacements at *z* = 0 (the mid-plane of a plate); $\phi_{x}$ and $\phi_{y}$ are the transverse normal rotations of the *y* and *x* axes; the comma denotes the differentiation with respect to *x* and *y* coordinates.

Four nodes per element, seven degrees of freedom per node were used for this problem. The displacement vector of node *i* for plate element is as follows:(4)$$q_{e}^{i}={\left\{u_{0i}\;v_{0i}\;w_{i}\;\phi_{xi}\;\phi_{yi}\;\frac{\partial w_{0i}}{\partial x}\;\frac{\partial w_{0i}}{\partial y}\right\}}^{T},\;i=1÷4$$

Because of the degree of freedom, *w* had the additional first derivative components $\frac{\partial w_{0i}}{\partial x},\;\frac{\partial w_{0i}}{\partial y}$. Therefore, in order to guarantee the continuous condition of displacement *w* and its first derivative components at each node, we had to approximate the displacement *w* by Hermite interpolation functions. The other four degrees of freedom were approximated by Lagrange interpolation functions.

The displacements of the plate in this approach may be expressed as follows:(5)$$\left\{u_{0},\;v_{0},\;\phi_{x},\;\phi_{y}\right\}=\left\{\sum_{i=1}^{4}N_{i}u_{0i},\;\sum_{i=1}^{4}N_{i}v_{0i},\;\sum_{i=1}^{4}N_{i}\phi_{xi},\;\sum_{i=1}^{4}N_{i}\phi_{yi}\right\}$$
(6)$$w=H_{1}w_{01}+H_{2}\frac{\partial w_{01}}{\partial x}+H_{3}\frac{\partial w_{01}}{\partial y}+\ldots +H_{10}w_{04}+H_{11}\frac{\partial w_{04}}{\partial x}+H_{12}\frac{\partial w_{04}}{\partial y}$$
(7)$$\frac{\partial w}{\partial x}=\frac{\partial }{\partial x}\left(H_{1}w_{01}+H_{2}\frac{\partial w_{01}}{\partial x}+H_{3}\frac{\partial w_{01}}{\partial y}+\ldots +H_{10}w_{04}+H_{11}\frac{\partial w_{04}}{\partial x}+H_{12}\frac{\partial w_{04}}{\partial y}\right)$$
(8)$$\frac{\partial w}{\partial y}=\frac{\partial }{\partial y}\left(H_{1}w_{01}+H_{2}\frac{\partial w_{01}}{\partial x}+H_{3}\frac{\partial w_{01}}{\partial y}+\ldots +H_{10}w_{04}+H_{11}\frac{\partial w_{04}}{\partial x}+H_{12}\frac{\partial w_{04}}{\partial y}\right)$$
where *N_i_* are Lagrange interpolating functions and *H_i_* are Hermite interpolating functions. 

The displacement vector is interpolated through the element’s nodal displacement vector as follows:(9)$$u_{0}=B_{H}\cdot q_{e}$$
where $B_{H}$ is the interpolation function matrix; $u_{0}$ and $q_{e}$ are expressed as follows:(10)$$u_{0}={\left\{u_{0},v_{0},w_{0},\phi_{x},\phi_{y},w_{0,x},w_{0,y}\right\}}^{T}$$
(11)$$q_{e}={\left\{q_{e}^{1}{\;q}_{e}^{2}{\;q}_{e}^{3}{\;q}_{e}^{4}\right\}}^{T}$$

The total strain of this plate in the case of the plate subjected to in-plane prebuckling stresses can be written as follows:(12)$$\left\{\begin{array}{l} \varepsilon \\ \gamma \end{array}\right\}=\left\{\begin{array}{l} \varepsilon^{\left(0\right)}\\ \gamma^{\left(0\right)} \end{array}\right\}+z\left\{\begin{array}{l} \varepsilon^{\left(1\right)}\\ 0 \end{array}\right\}+z^{2}\left\{\begin{array}{l} 0\\ \gamma^{\left(2\right)} \end{array}\right\}+z^{3}\left\{\begin{array}{l} \varepsilon^{\left(3\right)}\\ 0 \end{array}\right\}+\left\{\varepsilon_{P}^{G}\right\}$$
with
(13)$$\varepsilon^{\left(0\right)}=\left\{\begin{array}{l} \frac{\partial u_{0}}{\partial x}\\ \frac{\partial v_{0}}{\partial y}\\ \frac{\partial u_{0}}{\partial y}+\frac{\partial v_{0}}{\partial x} \end{array}\right\},\varepsilon^{\left(1\right)}=\frac{1}{4}\left\{\begin{array}{l} 5\left(\frac{\partial \phi_{x}}{\partial x}+\frac{\partial^{2}w}{\partial x^{2}}\right)\\ 5\left(\frac{\partial \phi_{y}}{\partial y}+\frac{\partial^{2}w}{\partial y^{2}}\right)\\ \left(\frac{\partial \phi_{x}}{\partial y}+2\frac{\partial^{2}w}{\partial x\partial y}+\frac{\partial \phi_{y}}{\partial x}\right) \end{array}\right\},\varepsilon^{\left(3\right)}=\frac{-5}{3h^{2}}\left\{\begin{array}{l} \frac{\partial \phi_{x}}{\partial x}+\frac{\partial^{2}w}{\partial x^{2}}\\ \frac{\partial \phi_{y}}{\partial y}+\frac{\partial^{2}w}{\partial y^{2}}\\ \frac{\partial \phi_{x}}{\partial y}+2\frac{\partial^{2}w}{\partial x\partial y}+\frac{\partial \phi_{y}}{\partial x} \end{array}\right\}$$
(14)$$\gamma^{\left(0\right)}=\frac{1}{4}\left\{\begin{array}{c} 5\phi_{x}+\frac{\partial w}{\partial x}\\ 5\phi_{y}+\frac{\partial w}{\partial y} \end{array}\right\};\gamma^{\left(2\right)}=\frac{-5}{h^{2}}\left\{\begin{array}{c} \phi_{x}+\frac{\partial w}{\partial x}\\ \phi_{y}+\frac{\partial w}{\partial y} \end{array}\right\};\varepsilon_{P}^{G}=\left\{\begin{array}{c} \frac{1}{2}{\left(\frac{\partial w}{\partial x}\right)}^{2}+\frac{z^{2}}{2}{\left(\frac{\partial \phi_{x}}{\partial x}\right)}^{2}+\frac{z^{2}}{2}{\left(\frac{\partial \phi_{y}}{\partial x}\right)}^{2}\\ \frac{1}{2}{\left(\frac{\partial w}{\partial y}\right)}^{2}+\frac{z^{2}}{2}{\left(\frac{\partial \phi_{x}}{\partial y}\right)}^{2}+\frac{z^{2}}{2}{\left(\frac{\partial \phi_{y}}{\partial y}\right)}^{2}\\ \frac{\partial w}{\partial x}\frac{\partial w}{\partial y}+z^{2}\frac{\partial \phi_{x}}{\partial x}\frac{\partial \phi_{x}}{\partial y}+z^{2}\frac{\partial \phi_{y}}{\partial x}\frac{\partial \phi_{y}}{\partial y}\\ 0\\ 0 \end{array}\right\}$$

By substituting Equation (4) into Equations (13) and (14), the strain field can be obtained as follows:(15)$$\varepsilon^{}=\left(B_{1}{+B}_{2}{+B}_{3}\right)q_{e};\gamma =\left(B_{4}{+B}_{5}\right)q_{e}$$
with
(16a)$$B_{1}=\sum_{i}^{4}\left[\begin{array}{ccccc} N_{i,x} & 0 & 0 & 0 & 0\\ 0 & N_{i,y} & 0 & 0 & 0\\ N_{i,y} & N_{i,x} & 0 & 0 & 0 \end{array}\right]$$
(16b)$$B_{2}=\frac{1}{4}\sum_{i=1}^{4}\left[\begin{array}{ccccccc} 0 & 0 & {\left(H_{\left(3i-2\right),x}\right)}_{,x} & 5N_{i,x} & 0 & {\left(H_{\left(3i-1\right),x}\right)}_{,x} & {\left(H_{\left(3i\right),x}\right)}_{,x}\\ 0 & 0 & {\left(H_{\left(3i-2\right),y}\right)}_{,y} & 0 & 5N_{i,y} & {\left(H_{\left(3i-1\right),y}\right)}_{,y} & {\left(H_{\left(3i\right),y}\right)}_{,y}\\ 0 & 0 & 2{\left(H_{\left(3i-2\right),x}\right)}_{,y} & 5N_{i,y} & 5N_{i,x} & 2{\left(H_{\left(3i-1\right),x}\right)}_{,y} & 2{\left(H_{\left(3i\right),x}\right)}_{,y} \end{array}\right]$$
(16c)$$B_{4}^{}=\frac{5}{4}\sum_{i=1}^{4}\left[\begin{array}{ccccccc} 0 & 0 & H_{\left(3i-2\right),x} & N_{i} & 0 & H_{\left(3i-1\right),x} & H_{3i,x}\\ 0 & 0 & H_{\left(3i-2\right),y} & 0 & N_{i} & H_{\left(3i-1\right),y} & H_{3i,y} \end{array}\right]$$
(16d)$$B_{5}^{}=-\frac{5}{h^{2}}\sum_{i=1}^{4}\left[\begin{array}{ccccccc} 0 & 0 & H_{\left(3i-2\right),x} & N_{i} & 0 & H_{\left(3i-1\right),x} & H_{3i,x}\\ 0 & 0 & H_{\left(3i-2\right),y} & 0 & N_{i} & H_{\left(3i-1\right),y} & H_{3i,y} \end{array}\right]$$

The constitutive relations are derived from Hooke’s law by the following equation:(17)$$\sigma =D_{m}\left(z\right)\left(\varepsilon^{\left(0\right)}+z\varepsilon^{\left(1\right)}+z^{3}\varepsilon^{\left(3\right)}\right);\;\tau =D_{s}\left(z\right)(\gamma^{\left(0\right)}+z^{2}\gamma^{\left(2\right)})$$
with
(18a)$$\sigma ={\left\{\sigma_{x},\sigma_{y},\tau_{xy}\right\}}^{T};\;\tau ={\left\{\tau_{xz},\tau_{yz}\right\}}^{T}$$
(18b)$$D_{m}\left(z\right)=\frac{E(z)}{1-v^{2}(z)}\left[\begin{array}{ccc} 1 & v(z) & 0\\ v(z) & 1 & 0\\ 0 & 0 & \left(1-v(z)\right)/2 \end{array}\right],\;D_{s}\left(z\right)=\frac{E(z)}{2\left(1+v(z)\right)}\left[\begin{array}{cc} 1 & 0\\ 0 & 1 \end{array}\right]$$

In this work, the stiffener was assumed to be parallel to the *x*-axis (See Figure 2). There was no delamilation phenomenon between the stiffener and the plate during the performance of the structure; the stiffener seemed to be a beam, and it just bent in the *zx*-plane. The displacement field of the *x*-stiffener can be expressed as follows:(19a)$$u^{xs}\left(x,y,z,t\right)=u_{0}^{xs}\left(x,y,t\right)+\frac{5}{4}\left(z-\frac{4}{3h_{xs}^{2}}z^{3}\right)\phi_{x}^{xs}\left(x,y,t\right)+\left(\frac{1}{4}z-\frac{5}{3h_{xs}^{2}}z^{3}\right)w_{0,x}^{xs}$$
(19b)$$v^{xs}\left(x,y,z,t\right)=0$$
(19c)$$w^{xs}\left(x,y,z,t\right)=w_{0}^{xs}\left(x,y,t\right)$$

The displacement vector of node *i* for the *x*-stiffener element is written as follows:(20)$$q_{i}^{xs}={\left\{u_{0i}^{xs}\;0\;w_{0i}^{xs}\;\phi_{xi}^{xs}\;0\;w_{0,x}^{xs}\;0\right\}}^{T},\;i=1–2$$

The displacements of the stiffener can now be given in the following form:(21)$${\left\{u_{0}^{xs},\;\phi_{x}^{xs}\right\}}^{T}={\left\{\sum_{i=1}^{2}N_{i}^{xs}u_{0i}^{xs},\sum_{i=1}^{2}N_{i}^{xs}\phi_{xi}^{xs}\right\}}^{T}$$
(22)$$w_{o}^{xs}=H_{1}^{xs}w_{01}^{xs}+H_{2}^{xs}{\left(\frac{\partial w}{\partial x}\right)}_{1}+H_{3}{\left(0\right)}_{1}+H_{4}^{xs}w_{02}^{xs}+H_{5}^{xs}{\left(\frac{\partial w}{\partial x}\right)}_{2}+H_{6}^{xs}{\left(0\right)}_{2}$$
(23)$$\frac{\partial w_{0}^{xs}}{\partial x}=\frac{\partial }{\partial x}\left(H_{1}^{xs}w_{01}^{xs}+\frac{\partial H_{2}^{xs}}{\partial x}{\left(\frac{\partial w_{0}^{xs}}{\partial x}\right)}_{1}+\frac{\partial H_{3}^{xs}}{\partial x}{\left(0\right)}_{1}+\frac{\partial H_{4}^{xs}}{\partial x}w_{02}^{xs}+\frac{\partial H_{5}^{xs}}{\partial x}{\left(\frac{\partial w_{0}^{xs}}{\partial x}\right)}_{2}+\frac{\partial H_{6}^{xs}}{\partial x}{\left(0\right)}_{2}\right)$$
where $N_{i}^{xs}$ are Lagrange interpolating functions, and $H_{i}^{xs}$ are Hermite interpolating functions. These functions can be obtained by substituting *s* = *s*_0_ into *N_i_* and *H_i_* of the plate.

The displacement vector of *x*-stiffener is interpolated as follows:(24)$$u_{0}^{xs}=B_{H}^{xs}{\cdot q}_{e}^{xs}$$
with
(25)$$u_{0}^{xs}={\left\{u_{0}^{xs},0,w_{0}^{xs},\phi_{x}^{xs},0,w_{0,x}^{xs},0\right\}}^{T}$$
(26)$$q_{e}^{\mathrm{xs}}={\left\{q_{1e}^{\mathrm{xs}}{\;q}_{2e}^{\mathrm{xs}}\right\}}^{T}$$

The strain components of the stiffener are as follows:(27)$$\left\{\begin{array}{l} \varepsilon_{xs}\\ \gamma_{xs} \end{array}\right\}=\left\{\begin{array}{l} \varepsilon_{xs}^{\left(0\right)}\\ \gamma_{xs}^{\left(0\right)} \end{array}\right\}+z\left\{\begin{array}{l} \varepsilon_{xs}^{\left(1\right)}\\ 0 \end{array}\right\}+z^{2}\left\{\begin{array}{l} 0\\ \gamma_{xs}^{\left(2\right)} \end{array}\right\}+z^{3}\left\{\begin{array}{l} \varepsilon_{xs}^{\left(3\right)}\\ 0 \end{array}\right\}+\left\{\varepsilon_{xs}^{G}\right\}$$
with
(28)$$\varepsilon_{xs}^{\left(0\right)}=\left\{\begin{array}{c} u_{0,x}^{xs}\\ 0\\ 0 \end{array}\right\};\;\varepsilon_{xs}^{\left(1\right)}=\frac{1}{4}\left\{\begin{array}{c} 5\left(\phi_{x,x}^{xs}+w_{,xx}^{xs}\right)\\ 0\\ 0 \end{array}\right\};\;\varepsilon_{xs}^{\left(3\right)}=\frac{-5}{3h_{xs}^{2}}\left\{\begin{array}{c} \phi_{x,x}^{xs}+w_{,xx}^{xs}\\ 0\\ 0 \end{array}\right\}$$
(29)$$\gamma_{xs}^{\left(0\right)}=\frac{1}{4}\left\{\begin{array}{c} 5\phi_{x}^{xs}+w_{,x}^{xs}\\ 0 \end{array}\right\};\gamma_{xs}^{\left(2\right)}=\frac{-5}{h_{xs}^{2}}\left\{\begin{array}{c} \phi_{x}^{xs}+w_{,x}^{xs}\\ 0 \end{array}\right\};\varepsilon_{xs}^{G}=\left\{\begin{array}{c} \frac{1}{2}{\left(w_{,x}^{xs}\right)}^{2}+\frac{z^{2}}{2}{\left(\phi_{x,x}^{xs}\right)}^{2}\\ 0\\ 0\\ 0\\ 0 \end{array}\right\}$$

By substituting Equation (24) into Equations (27)–(29), the strain field is as follows:(30)$$\varepsilon_{xs}=\left(B_{1}^{xs}{+B}_{2}^{xs}{+B}_{3}^{xs}\right)q_{e}^{xs};\gamma_{xs}=\left(B_{4}^{xs}{+B}_{5}^{xs}\right)q_{e}^{xs}$$
with
(31a)$$B_{1}^{xs}=\sum_{i}^{2}\left[\begin{array}{cccc} N_{i,x}^{xs} & 0 & 0 & 0\\ 0 & 0 & 0 & 0\\ 0 & 0 & 0 & 0 \end{array}\begin{array}{c} 0\\ 0\\ 0 \end{array}\right]$$
(31b)$$B_{2}^{xs}=\frac{1}{4}\sum_{i=1}^{2}\left[\begin{array}{ccccccc} 0 & 0 & H_{\left(3i-2\right),xx}^{xs} & 5N_{i,x}^{xs} & 0 & H_{\left(3i-1\right),xx}^{xs} & H_{\left(3i\right),xx}^{xs}\\ 0 & 0 & 0 & 0 & 0 & 0 & 0\\ 0 & 0 & 0 & 0 & 0 & 0 & 0 \end{array}\right]$$
(31c)$$B_{3}^{xs}=-\frac{5}{3h_{xs}^{2}}\sum_{i=1}^{2}\left[\begin{array}{ccccccc} 0 & 0 & H_{\left(3i-2\right),xx}^{xs} & N_{i,x}^{xs} & 0 & H_{\left(3i-1\right),xx}^{xs} & H_{\left(3i\right),xx}^{xs}\\ 0 & 0 & 0 & 0 & 0 & 0 & 0\\ 0 & 0 & 0 & 0 & 0 & 0 & 0 \end{array}\right]$$
(31d)$$B_{4}^{xs}=\frac{5}{4}\sum_{i=1}^{2}\left[\begin{array}{ccccccc} 0 & 0 & H_{\left(3i-2\right),x}^{xs} & N_{i}^{xs} & 0 & H_{\left(3i-1\right),x}^{xs} & H_{\left(3i\right),x}^{xs}\\ 0 & 0 & 0 & 0 & 0 & 0 & 0 \end{array}\right]\;$$
(31e)$$B_{5}^{xs}=-\frac{5}{h_{xs}^{2}}\left[\begin{array}{ccccccc} 0 & 0 & H_{\left(3i-2\right),x}^{xs} & N_{i}^{xs} & 0 & H_{\left(3i-1\right),x}^{xs} & H_{\left(3i\right),x}^{xs}\\ 0 & 0 & 0 & 0 & 0 & 0 & 0 \end{array}\right]$$

The relationship between stresses and strains obtained from Hooke’s law is as follows:(32)$$\sigma_{xs}^{}=D_{m}^{xs}\left(\varepsilon_{xs}^{\left(0\right)}+z.\varepsilon_{xs}^{\left(1\right)}+z^{3}\varepsilon_{xs}^{\left(3\right)}\right);\tau_{xs}=D_{s}^{xs}\left(\gamma_{xs}^{\left(0\right)}+z^{2}\gamma_{xs}^{\left(2\right)}\right)$$
with
(33a)$$\sigma_{xs}={\left[\begin{array}{ccc} \sigma_{x}^{xs} & 0 & 0 \end{array}\right]}^{T},\tau_{xs}={\left[\begin{array}{cc} \tau_{xz}^{xs} & 0 \end{array}\right]}^{T}$$
(33b)$$D_{m}^{xs}=\frac{E_{xs}}{1-v_{xs}^{2}}\left[\begin{array}{ccc} 1 & v_{xs} & 0\\ v_{xs} & 1 & 0\\ 0 & 0 & \left(1-v_{xs}\right)/2 \end{array}\right]$$
(33c)$$D_{s}^{xs}=\frac{E_{xs}}{2\left(1+v_{xs}\right)}\left[\begin{array}{cc} 1 & 0\\ 0 & 1 \end{array}\right]$$

The *x*-stiffener is considered to place at the lower surface of the plate. The condition of displacement at the contact line is as follows:(34)$$u_{0|z=-0.5h}=u_{0|z=-0.5h_{xs}}^{xs}$$

Using Equations (3) and (19), Equation (34) becomes the following:(35)$$\left\{ \begin{array}{l} u_{0i}^{xs}=u_{0i}^{}+e_{1}^{xs}\phi_{xi}+e_{2}^{xs}w_{0i,x}\\ \phi_{xi}^{xs}=\phi_{xi}^{};w_{0i,x}^{xs}=w_{0i,x}^{} \end{array}\right.$$
where
(36)$$e_{1}^{xs}=-\frac{5}{12}\left(h+h_{xs}\right);e_{2}^{xs}=\frac{1}{12}\left(h+h_{xs}\right)$$

Equation (35) can be rewritten as follows:(37)$$\left\{\begin{array}{l} u_{0i}^{xs}\\ 0\\ w_{0i}^{xs}\\ \phi_{xi}^{xs}\\ 0\\ w_{0i,x}^{xs}\\ 0 \end{array}\right\}=\left[\begin{array}{ccccccc} 1 & 0 & 0 & e_{1}^{xs} & 0 & e_{2}^{xs} & 0\\ 0 & 0 & 0 & 0 & 0 & 0 & 0\\ 0 & 0 & 1 & 0 & 0 & 0 & 0\\ 0 & 0 & 0 & 1 & 0 & 0 & 0\\ 0 & 0 & 0 & 0 & 0 & 0 & 0\\ 0 & 0 & 0 & 0 & 0 & 1 & 0\\ 0 & 0 & 0 & 0 & 0 & 0 & 0 \end{array}\right]\left\{\begin{array}{l} u_{0i}^{}\\ v_{0i}^{}\\ w_{0i}^{}\\ \phi_{xi}^{}\\ \phi_{yi}^{}\\ w_{0i,x}^{}\\ w_{0i,y}^{} \end{array}\right\}$$
or in shorter form:(38)$$u_{0i}^{xs}=T_{x}{\cdot u}_{0i}^{}$$

The nodal displacement vector of stiffener element is as follows:(39)$$q_{e}^{xs}=T_{xs}{\cdot q}_{e}^{}$$
in which
(40)$$T_{xs}{=T}_{x}diag(4,4)$$

The elastic strain energy of the stiffened plate is written as follows:(41)$$U=\frac{1}{2}\sum_{}^{N_{p}}\int_{V_{e}}\varepsilon^{T}\cdot \sigma \cdot dV+\frac{1}{2}\sum_{}^{N_{xs}}\int_{V_{e}^{xs}}\varepsilon_{xs}^{T}\cdot \sigma_{xs}\cdot dV$$
or in matrix form:(42)$$U=\frac{1}{2}\sum_{}^{N_{p}}q_{e}^{T}K_{e}^{p}q_{e}^{}+\frac{1}{2}\sum_{}^{N_{xs}}{\left(q_{e}^{}\right)}^{T}K_{e}^{xs}q_{e}^{}$$
where
(43)$$K_{e}^{p}=\int_{S_{e}}\left[\begin{array}{l} B_{1}^{T}{\mathrm{AB}}_{1}{+B}_{1}^{T}{\mathrm{BB}}_{2}{+B}_{1}^{T}{\mathrm{EB}}_{3}{+B}_{2}^{T}{\mathrm{BB}}_{1}{+B}_{2}^{T}{\mathrm{DB}}_{2}{+B}_{2}^{T}{\mathrm{FB}}_{3}{+B}_{3}^{T}{\mathrm{EB}}_{1}+\\ {+B}_{3}^{T}{\mathrm{FB}}_{2}{+B}_{3}^{T}{\mathrm{HB}}_{3}{+B}_{4}^{T}{A^{\prime }B}_{4}{+B}_{4}^{T}{B^{\prime }B}_{5}{+B}_{5}^{T}{B^{\prime }B}_{4}{+B}_{5}^{T}{D^{\prime }B}_{5} \end{array}\right]dS$$
(44)$$K_{e}^{xs}=T_{xs}^{T}\left(b_{xs}\int_{l_{e}^{xs}}\left[\begin{array}{l} {\left(B_{1}^{xs}\right)}^{T}A^{xs}B_{1}^{xs}+{\left(B_{1}^{xs}\right)}^{T}B^{xs}B_{2}^{xs}+{\left(B_{1}^{xs}\right)}^{T}E^{xs}B_{3}^{xs}+{\left(B_{2}^{xs}\right)}^{T}B^{xs}B_{1}^{xs}+\\ +{\left(B_{2}^{xs}\right)}^{T}{\;\tilde{D}}^{xs}B_{2}^{xs}+{\left(B_{2}^{xs}\right)}^{T}F^{xs}B_{3}^{xs}+{\left(B_{3}^{xs}\right)}^{T}E^{xs}B_{1}^{xs}+{\left(B_{3}^{xs}\right)}^{T}F^{xs}B_{2}^{xs}+\\ +{\left(B_{3}^{xs}\right)}^{T}H^{xs}B_{3}^{xs}+{\left(B_{4}^{xs}\right)}^{T}A_{xs}^{\prime }B_{4}^{xs}+{\left(B_{4}^{xs}\right)}^{T}B_{xs}^{\prime }B_{5}^{xs}+\\ +{\left(B_{5}^{xs}\right)}^{T}B_{xs}^{\prime }B_{4}^{xs}+{\left(B_{5}^{xs}\right)}^{T}D_{xs}^{\prime }B_{5}^{xs} \end{array}\right]dl\right)T_{xs}^{}$$
in which

(45)$$\left(A,\;B,\;D,\;E,\;F,\;H\right)=\int_{-h/2}^{h/2}\left(1,\;z,\;z^{2},\;z^{3},\;z^{4},\;z^{6}\right)D_{m}\cdot dz$$

(46)$$\left(A^{\prime },\;B^{\prime },\;D^{\prime }\right)=\int_{-h/2}^{h/2}\left(1,z^{2},\;z^{4}\right)D_{s}\cdot dz$$

(47)$$\left(A^{xs},{\;B}^{xs},{\;\tilde{D}}^{xs},{\;E}^{xs},{\;F}^{xs},{\;H}^{xs}\right)=\int_{-h_{xs}/2}^{h_{xs}/2}\left(1,\;z,\;z^{2},\;z^{3},\;z^{4},\;z^{6}\right)D_{m}^{xs}\cdot dz$$

(48)$$\left(A_{xs}^{\prime },{\;B}_{xs}^{\prime },{\;D}_{xs}^{\prime }\right)=\int_{-h_{xs}/2}^{h_{xs}/2}\left(1,z^{2},\;z^{4}\right)D_{s}^{xs}\cdot dz$$

The geometric strain energy enforced by in-plane prebuckling stresses is then computed by the following:(49)$$U^{G}=\frac{1}{2}\sum_{}^{N_{p}}\int_{V_{e}}{\;\hat{\sigma }}_{0}^{T}\cdot \varepsilon_{P}^{G}\cdot dV+\frac{1}{2}\sum_{}^{N_{xs}}\int_{V_{e}^{xs}}{\;\hat{\sigma }}_{0xs}^{T}\cdot \varepsilon_{xs}^{G}\cdot dV$$

By substituting the geometric strain of the plate and the stiffeners into Equation (49), we get the following:(50)$$U^{G}=\frac{1}{2}\sum_{}^{N_{p}}\int_{S_{e}}{\left({\;\bar{\varepsilon }}_{P}^{G}\right)}^{T}\sigma_{0}^{}{\cdot \;\bar{\varepsilon }}_{P}^{G}\cdot dS+\frac{1}{2}\sum_{}^{N_{xs}}\int_{S_{e}^{xs}}{\left({\;\bar{\varepsilon }}_{xs}^{G}\right)}^{T}\sigma_{0}^{xs}{\;\bar{\varepsilon }}_{xs}^{G}dS$$
where
(51)$${\;\bar{\varepsilon }}_{P}^{G}=\left\{\begin{array}{c} w_{0,x}\\ w_{0,y}\\ \phi_{x,x}\\ \phi_{x,y}\\ \phi_{y,x}\\ \phi_{y,y} \end{array}\right\}=\left[\begin{array}{ccccccc} 0 & 0 & 0 & 0 & 0 & 1 & 0\\ 0 & 0 & 0 & 0 & 0 & 0 & 1\\ 0 & 0 & 0 & \frac{\partial }{\partial x} & 0 & 0 & 0\\ 0 & 0 & 0 & \frac{\partial }{\partial y} & 0 & 0 & 0\\ 0 & 0 & 0 & 0 & \frac{\partial }{\partial x} & 0 & 0\\ 0 & 0 & 0 & 0 & \frac{\partial }{\partial y} & 0 & 0 \end{array}\right]\left\{\begin{array}{c} u\\ v\\ w\\ \phi_{x}\\ \phi_{y}\\ w_{0,x}\\ w_{0,y} \end{array}\right\}=L_{P}^{G}q_{e}$$
(52)$$\sigma_{0}^{}=diag\left(h\;\tilde{\sigma },\;\frac{h^{3}}{12}\;\tilde{\sigma },\;\frac{h^{3}}{12}\;\tilde{\sigma }\right);\;\tilde{\sigma }={\left[\sigma_{x}^{0}\;\tau_{xy}^{0};\tau_{xy}^{0}\;\sigma_{y}^{0}\right]}^{T}$$
(53)$${\bar{\varepsilon }}_{xs}^{G}=\left\{\begin{array}{c} w_{0,x}\\ \phi_{x,x} \end{array}\right\}=\left[\begin{array}{ccccccc} 0 & 0 & 0 & 0 & 0 & 1 & 0\\ 0 & 0 & 0 & \partial /\partial x & 0 & 0 & 0 \end{array}\right]\left\{\begin{array}{c} u^{xs}\\ 0\\ w_{0}^{xs}\\ \phi_{x}^{xs}\\ 0\\ w_{0,x}^{xs}\\ 0 \end{array}\right\}=L_{xs}^{G}q_{e}^{xs}$$
(54)$$\sigma_{0}^{xs}=diag\left(h_{xs}^{}{\tilde{\sigma }}^{xs},\;\frac{h_{xs}^{3}}{12}{\tilde{\sigma }}^{xs},\;\frac{h_{xs}^{3}}{12}{\tilde{\sigma }}^{xs}\right);{\tilde{\sigma }}^{xs}=\left[\sigma_{x}^{0}\;0;0\;0\right]$$

Equation (50) now becomes the following:(55)$$U^{G}=\frac{1}{2}\sum_{}^{N_{p}}q_{e}^{T}K_{e}^{Gp}q_{e}^{}+\frac{1}{2}\sum_{}^{N_{xs}}{\left(q_{e}^{}\right)}^{T}K_{e}^{Gxs}q_{e}^{}$$
with
(56)$$K_{e}^{Gp}=\int_{S_{e}}{\left(L_{p}^{G}\right)}^{T}\sigma_{0}L_{p}^{G}dS;{\;K}_{e}^{Gxs}=T_{xs}^{T}\left(b_{xs}\int_{l_{e}}{\left(L_{xs}^{G}\right)}^{T}\sigma_{0}^{xs}L_{xs}^{G}dl\right)T_{xs}^{}$$

For the buckling problem, we get the following equation:(57)$$\left\{\left(K^{p}+K_{}^{\mathrm{xs}}\right)-\lambda_{b}\left(K^{\mathrm{Gp}}+K_{}^{\mathrm{Gxs}}\right)\right\}d=\;0\;$$
where $K^{p},{\;K}_{}^{xs}$ and $K^{Gp},{\;K}_{}^{Gxs}$ are the global stiffness matrix and global geometric stiffness matrix, respectively. d stands for the vector of unknowns. Equation (57) is solved to obtain the buckling load $\lambda_{b}$ and the buckling mode shape.

## 3. Numerical Results

### 3.1. Formulation Verification

First, we conducted a comparison of the critical buckling loads for a simply supported FGM plate with the analytical results of Meisam et al. [27] and Bodaghi et al. [28], as shown in Table 1. We considered a square plate with *a* = *b* = 1 m, the thickness *h* = *a*/100, the material properties *E_c_* = 380 × 10^9^ Pa, *E_m_* = 70 × 10^9^ Pa, the Poisson ratio 0.3, and the plate as being under an axial compression at two opposite edges.

Next, we compared the buckling coefficient of the clamped plate with one central stiffener with a model, as shown in Figure 3. The plate had geometrical parameters *a*/*b* = 1 and thickness of *h* = *a*/200; the depth of the stiffener was *h_s_* = 10.483*h*, and the width was *b_s_* = *h_s_*/2.75. The material properties of the plate and the stiffener were *E* = 68.7 GPa, ν = 0.3. The buckling coefficient was compared with the results of Mukhopadhyay et al. [29] (semianalytical finite difference method), Peng et al. [30] (mesh-free method), and Rikards et al. [31] (finite element method). This comparison is listed in Table 2.
(58)$$k_{buk}=\lambda_{b}\cdot a_{}^{2}/(D\cdot \pi_{}^{3})$$
with $D=Eh^{3}/12(1-\nu^{2})$. 

From Table 1 and Table 2, we can see clearly that the result of our work compared with other approaches has a very small error, so the calculation program used in this paper is verified. 

### 3.2. Buckling of Rectangular FGM Plate with One Central Stiffener

#### 3.2.1. Long FGM Plate with One Central Stiffener

In this work, two types of FGM plates (Si_3_N_4_/SUS304 plate, ZrO_2_/SUS304 plate) were employed. The material properties are presented in Table 3. We considered a rectangular plate with one central stiffener with *b* = 0.2 m, *a*/*b* = 2.5. Here, the stiffener was made of the same material as the material of the plate. 

The buckling coefficient is calculated as follows:(59)$$k_{buk}=\lambda_{buk}\cdot a_{0}^{2}/\left(E_{0}\cdot h_{0}^{3}\right)$$
where *E*_0_ = 5 × 10^7^ Pa, *a*_0_ = 10*h*_0_. 


*-Effect of the Depth of Stiffener (h_s_)*


We evaluated how the depth of the stiffener affects the buckling behavior of the stiffened FGM plate. The volume fraction index *n* had a value in the range from 0.1 to 10. Considering three cases of plates with *b*/*h* = 100, 150, and 200, the geometrical parameters of the stiffener were *b_s_* = 2*h*, *h_s_* = *h* − 4*h*; the plate was fully simply supported (SSSS). We also determined the mass reduction of the stiffened structure compared with the plate without stiffeners with the same dimensions (the length and the width of this plate). For the stiffened plate with the thickness *h* = *b*/100, the buckling coefficients are represented by the horizontal dash-dot lines in the Figure 4. For each of the plates without stiffeners, we let the thickness of the plate change from *b*/95 to *b*/70. The buckling coefficients are represented by the solid lines in Figure 4 and are listed in Table 4. The thickness relationship between the Si_3_N_4_/SUS304 plate with and without a central stiffener is listed in Table 5 and Table 6. The red point, which is the intersection point between the dash-dot line and the solid line, represents the equivalent buckling coefficient of the plate without stiffeners and the stiffened plate. Following the same procedure for the stiffened plate with the thickness *h* = *b*/150 (the thickness of the unstiffened plate changed from *b*/140 to *b*/110) and *h* = *b*/200 (the thickness of the unstiffened plate changed from *b*/180 to *h* = *b*/155), we obtained the results given in Figure 5 and Table 7, Table 8, Table 9, Table 10, Table 11, Table 12, Table 13, Table 14, Table 15 and Table 16.

From Figure 4 and Figure 5 and Table 4, Table 5, Table 6, Table 7, Table 8, Table 9, Table 10, Table 11, Table 12, Table 13, Table 14, Table 15 and Table 16, we can come to the following conclusions:-The depth of stiffener (*h_s_*) has a strong influence on the buckling coefficient of the structure. For both Si_3_N_4_/SUS304 and ZrO_2_/SUS304 plates, when *h_s_* increased, the buckling coefficient increased, as we can see from the horizontal lines in the above figures.-For the Si_3_N_4_/SUS304 plate, with the three thicknesses *h* considered (*h* = *b*/100, *h* = *b*/150, *h* = *b*/200), when *n* was small, the addition of the stiffener enhanced the mass of the structure compared with the plate without stiffeners. In contrast, when *n* was large, the addition of stiffener significantly reduced the mass of the structure in order to obtain the same buckling coefficient between the stiffened plate (thinner plate) and the plate without a central stiffener (thicker plate). The maximum mass reduction was up to 13.17%, 18.16%, and 20% for cases *h* = *b*/200, *h* = *b*/150, *h* = *b*/100, respectively, herein *h_s_* = 4*h*.-For the ZrO_2_/SUS304 plate, for all values of *n*, the addition of stiffener decreased the mass of the structure compared with the unstiffened plate (thicker plate), but they still had the same buckling load. The larger the volume index fraction *n* got, the larger the mass reduction reached. The maximum mass reduction was up to 15.07%, 20.74%, and 23.26% for cases *h* = *b*/200, *h* = *b*/150, *h* = *b*/100, respectively, herein *h_s_* = 4*h*.

These numerical results are very interesting as it obviously demonstrates that the addition of stiffener still ensured the stability of the structure under a compressive load and that it could significantly reduce the mass of the structure due to the thickness of the plate decreasing. This point is really significant for many cases in practical engineering, where the mass of the structure must be reduced to make sure the process works under compressive loads, especially in aerospace engineering, shipbuilding and so on.


*-Effect of the gradient index (n)*


Next, the authors evaluated the influence of the gradient index *n* on the buckling response of the plate. We considered two cases of plates made of Si_3_N_4_/SUS304 or ZrO_2_/SUS304 with geometrical parameters *h_s_* = 2*h*, *b_s_* = *b*/100 and *b*/*h* ratio value of 10, 20, 50, and 100; the plate was fully simply supported (SSSS). By changing the volume fraction index *n* from 0.1 to 10, we had a variation of buckling coefficients of FGM plates with one central stiffener and without stiffeners, as shown in Figure 6.

Note that in Figure 6, the solid lines and dash-dot lines represent buckling coefficients of stiffened and unstiffened plates, respectively. The solid lines are always higher than the dash-dot lines, meaning that the buckling load of the stiffened plate was larger than that of the unstiffened plate. This proves that the stiffness of the plate will increase by the addition of the stiffener. We can see that when the thickness of the plate increased with a central stiffener, the plate became stiffer, leading to an enhancing of the buckling load of the plate. When the volume fraction index *n* was increased, the buckling load of the ZrO_2_/SUS304 plate got larger; in contrast, the buckling load of the Si_3_N_4_/SUS304 plate got smaller.

Figure 7 presents the first four buckling mode shapes of the FGM plate with and without a central stiffener. The figures show clearly that the stiffener had a strong influence on the buckling mode shape of the plate.


*-Effect of the length-to-width ratio (a/b)*


Next, we examined the effects of the length-to-width ratio (*a*/*b*) on the buckling load of the plate with one central stiffener. In this case, we constantly maintained the width of the plate as *b* = 0.2 m and the width of the stiffener as *b_s_* = *b*/100; the plate was under fully clamped (CCCC). By changing the length of the plate *a*, the *a*/*b* ratio reached the value from 2 to 4. We then obtained the diagram of the buckling load, as shown in Figure 8. The figure shows that when the length-to-width ratio (*a*/*b*) increased, the plate became “softer”, meaning the buckling load decreased. 

Figure 9 presents the first four buckling mode shapes of the stiffened plate with *a*/*b* = 2 and *a*/*b* = 4. The results demonstrate that the length and width of the plate strongly affected the buckling mode shape of the structure.


*-Effect of the boundary condition*


Next, we investigated the influence of the boundary condition on the buckling load of the stiffened FGM plate. Figure 10 and Figure 11 present the buckling coefficient of the structure. The results in Figure 10 were calculated for plates with various *b*/*h* ratios, and the results in Figure 11 were calculated for plates with various *a*/*b* ratios. From these figures, we can see that the fully clamped supported plate had the largest buckling load, and the fully simply supported plate had the smallest buckling load.

#### 3.2.2. Long FGM Plate with Two Stiffeners

We considered a rectangular plate with two parallel stiffeners, as shown in Figure 12, with *b* = 0.2 m, *a*/*b* = 2.5. The geometrical parameters of the stiffener were *h_s_* = *2h*, *b_s_* = *b*/100. The buckling coefficient of the plate can be determined by the following formula:(60)$$k_{buk}=\lambda_{buk}a_{0}^{2}/(E_{0}h_{0}^{3})$$
with *E*_0_ = 5 × 10^7^ Pa, *a*_0_ = 10*h*_0_. 


*-Effect of the width of the stiffener and the distance between two stiffeners*


The effects of the width of the stiffener and the distance between two stiffeners were studied. We compared the buckling loads between the plate with two stiffeners (*b_s_* = *b*/100, *b_s_* = *b*/200, Figure 13b) and the one with one central stiffener (*b_s_* = *b*/100, Figure 13a). 

By varying the volume index fraction *n* and the distance between the two stiffeners *d*, we obtained the numerical results of the buckling coefficients of the plate, as plotted in Figure 14.

From Figure 14, we can reach the following conclusions. First, Figure 14 clearly shows that the distance between the two stiffeners strongly influences the buckling coefficient of the structure. In both cases, i.e., Si_3_N_4_/SUS304 and ZrO_2_/SUS304 plates, when *d* increased, the buckling coefficient of the plate decreased. This is explained by the fact that the strain energy focuses on the central area of the structure. Therefore, when the stiffeners are set there, the stiffness of this plate is larger than other places, thus leading to enhanced buckling coefficient of the plate.

Second, similar to the previous sections, when the volume index fraction *n* increased, the buckling coefficient of the Si_3_N_4_/SUS304 plate decreased, and the buckling coefficient of the ZrO_2_/SUS304 plate increased.

Third, we explored the different buckling coefficients between a plate with a central stiffener and a plate with two stiffeners. We considered a plate with two stiffeners and a plate with a central stiffener with the same geometrical parameters except the width of the stiffener *b_s_*. In case 1, we considered a plate with two stiffeners, which had the total *b_s_* of two stiffeners (each stiffener had *b_s_*/2) equal to the *b_s_* of the plate with a central stiffener (Figure 13a,b). For both computed results of Si_3_N_4_/SUS304 and ZrO_2_/SUS304 plates (Figure 14a,b), the buckling coefficient of the plate with one central stiffener was always larger than the buckling coefficient of the plate with two stiffeners. When the distance *d* was small, the plate with two stiffeners had the same buckling coefficient as a plate with one central stiffener. In case 2, we considered a plate with two stiffeners, with the width of each stiffener *b_s_* equal to *b_s_* of the plate with a central stiffener (Figure 13a,c). As can be seen in Figure 14b,d, for both Si_3_N_4_/SUS304 and ZrO_2_/SUS304 plates, when *d* = 0.5*b*, the buckling coefficient of the structure was the same as the plate with one stiffener. When the distance of stiffeners was *d* < 0.5*b*, the buckling coefficient of the plate with two stiffeners was larger than that of the plate with a central stiffener. When *d* > 0.5*b*, the buckling coefficient of the plate with two stiffeners was smaller than that of the plate with a central stiffener. These interesting results have a good significance in practical engineering. For case 1, all plates had the same volume of the stiffener, but using the plate with one central stiffener was much better due to its buckling load always being higher than that of the plate with two stiffeners. In case 2, the plate with two stiffeners was only better than that with one stiffener in the buckling problem when the distance between the two stiffeners was suitable. 


*-Effect of the plate thickness*


In this study, the thickness of the plate (the plate model is given in Figure 12) had values from *b*/50 to *b*/5, and the numerical results are shown in Figure 15. We can again see the same phenomenon as in the case of the plate with a central stiffener. When the volume fraction index *n* increased, the buckling coefficient of Si_3_N_4_/SUS304 plate increased, and the buckling coefficient of ZrO_2_/SUS304 plate decreased. The reason for this is that the plate became softer.

Figure 16 presents the first four buckling mode shape of the plate with *b*/*h* = 5 and *b*/*h* = 50. We can see that the thickness of the structure strongly affected the buckling loads as well as buckling mode shapes.


*-Effect of the length-to-width ratio (a/b)*


We next explored the effects of the length-to-width ratio (*a*/*b*). The geometrical parameters were *b* = 0.2 m, *h* = *b*/10. We changed the length *a* and the volume fraction index *n*, and the results are plotted in Figure 17. As can be seen from the figure, the length-to-width ratio *(a*/*b)* had a robust influence on the buckling coefficient of the structure. This point can be explained by the fact that when the *a*/*b* ratio decreased, the length *a* correspondingly increased. This led to the plate becoming “softer”, and the buckling coefficient was therefore reduced.

Figure 18 presents the first four buckling mode shapes of the FGM plate for two cases, i.e., *a*/*b* = 2 and *a*/*b* = 5; the plate is fully simply supported (SSSS). We can see that both the boundary condition and the *a*/*b* ratio strongly affected not only the buckling loads but also the buckling mode shapes of the structure.

## 4. Conclusions

In this study, a linear finite element formulation based on the Shi shear deformation theory was employed to study the linear mechanical buckling of stiffened functionally graded material plates. Different parameters were examined, including the geometrical and mechanical properties of the plate. Based on the numerical results presented in the above sections, some major conclusions are listed as follows:-The volume index fraction *n* has a strong influence on the buckling load of the structure. Due to the different type of material, the buckling coefficient of the Si_3_N_4_/SUS304 plate decreased as the volume index fraction *n* increased. In contrast, the buckling coefficient of the ZrO_2_/SUS304 plate increased as the volume index fraction *n* increased.-The geometrical parameters of both the plate and the stiffener also strongly affect the buckling coefficient of the structure. For the Si_3_N_4_/SUS304 and ZrO_2_/SUS304 plates with one central stiffener or two stiffeners, the buckling load increased as the thickness of the stiffener increased.-The addition of stiffener to the plate can significantly reduce the total mass of the structure, but it still maintains the stability of the plate with the same buckling load as the plate without stiffeners. The maximum mass reduction could be reached at 20% for the Si_3_N_4_/SUS304 plate and 23.26% for the ZrO_2_/SUS304 plate. This point is very significant in practical engineering when we need to have lighter structures. Moreover, using a plate with two stiffeners was better than using a plate with one central stiffener when we had the right distance (*d* < 0.5*b*) between the two stiffeners.-Boundary conditions also deeply affect the buckling load of the structure. The plate with CCCC boundary condition had much more buckling load than the others.

## Figures and Tables

**Figure 1 materials-12-01262-f001:**
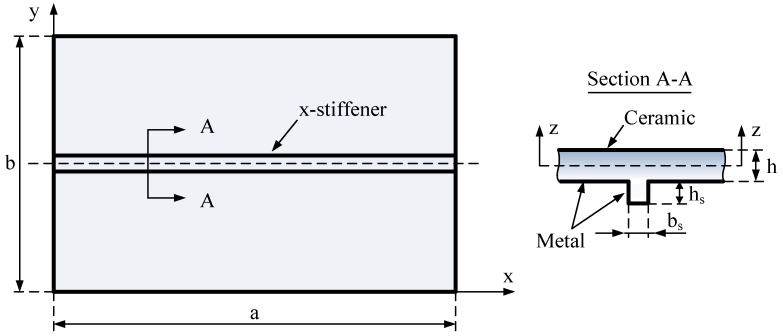
A functionally graded material (FGM) plate stiffened by an *x*-direction stiffener.

**Figure 2 materials-12-01262-f002:**
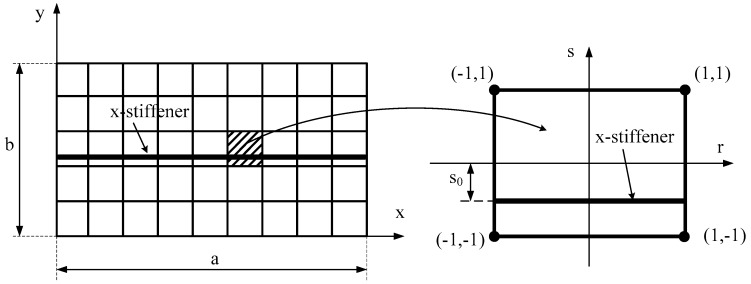
Plate element and stiffener element.

**Figure 3 materials-12-01262-f003:**
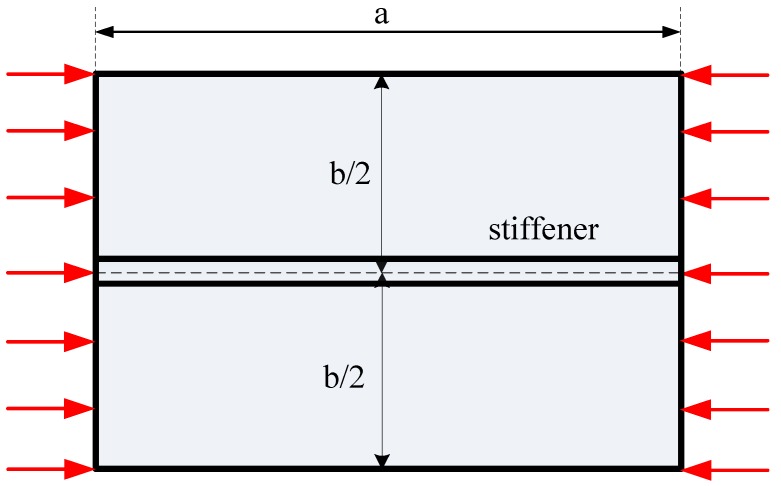
FGM plate with one central stiffener.

**Figure 4 materials-12-01262-f004:**
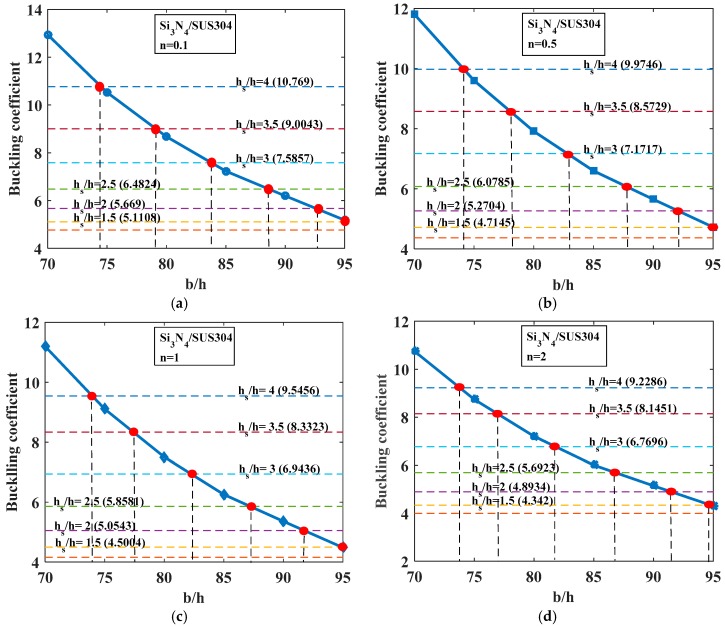
Buckling coefficient of FGM (Si_3_N_4_/SUS304) plate. Solid blue line: buckling coefficient of plate without stiffeners (*b*/*h* = 70–95); horizonal dash-dot line: buckling coefficient of plate with a central stiffener (*b*/*h* = 100) with various *h_s_*/*h* ratios. (**a**) *n* = 0.1; (**b**) *n* = 0.5; (**c**) *n* = 1; (**d**) *n* = 2; (**e**) *n* = 5; (**f**) *n* = 10.

**Figure 5 materials-12-01262-f005:**
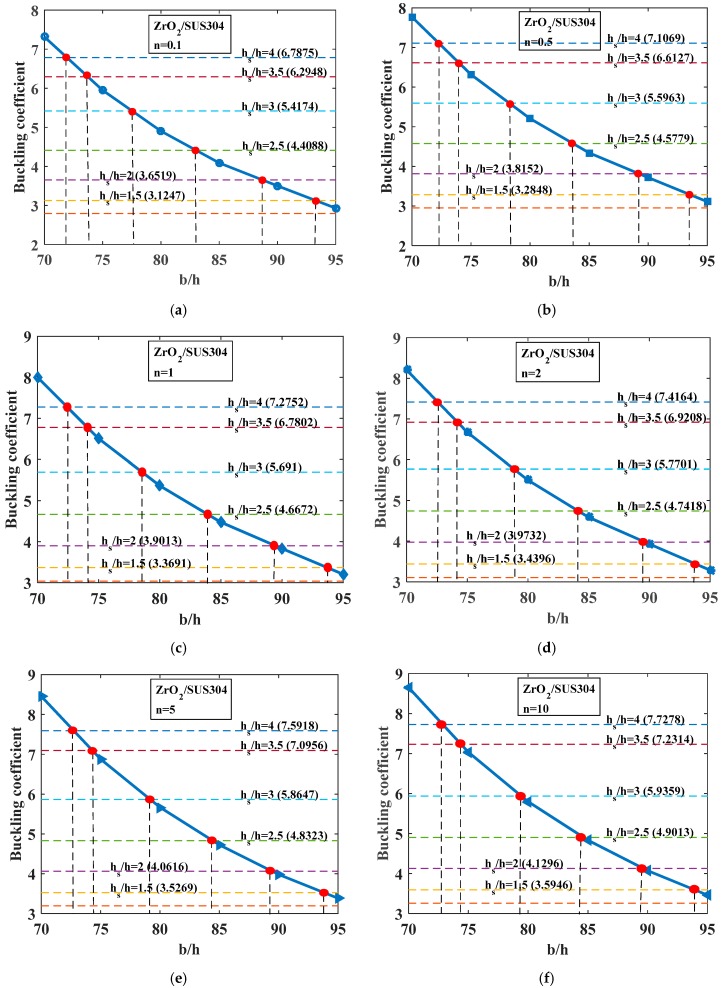
Buckling coefficient of FGM (ZrO_2_/SUS304) plate. Solid blue line: buckling coefficient of plate without stiffeners (*b*/*h* = 70–95); horizontal dash-dot line: buckling coefficient of plate that has central stiffener (*b*/*h* = 100) with various *h_s_*/*h* ratios. (**a**) *n* = 0.1; (**b**) *n* = 0.5; (**c**) *n* = 1; (**d**) *n* = 2; (**e**) *n* = 5; (**f**) *n* = 10.

**Figure 6 materials-12-01262-f006:**
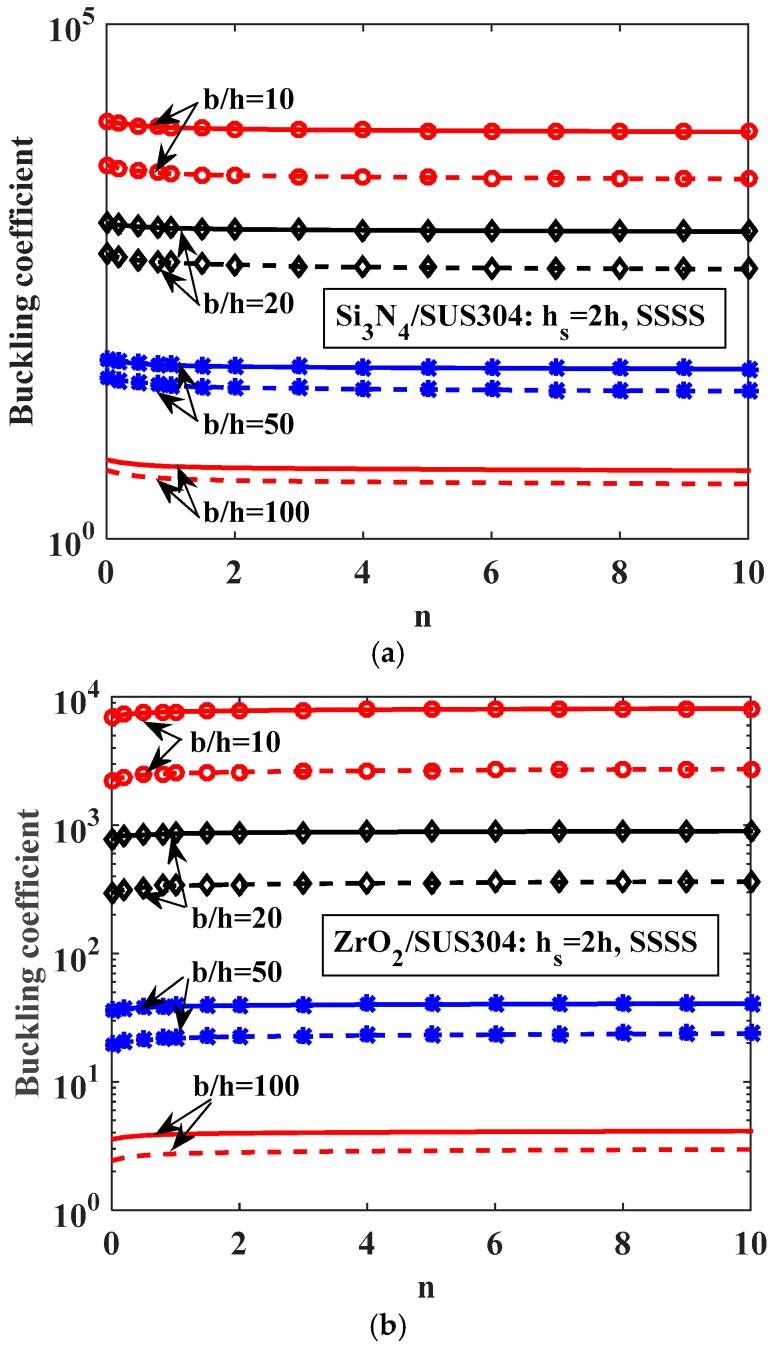
Variation of buckling coefficient of FGM plate with one central stiffener and without stiffeners (solid line: plate with one central stiffener, dash-dot line: plate without stiffeners). (**a**) Si_3_N_4_/SUS304; (**b**) ZrO_2_/SUS304.

**Figure 7 materials-12-01262-f007:**
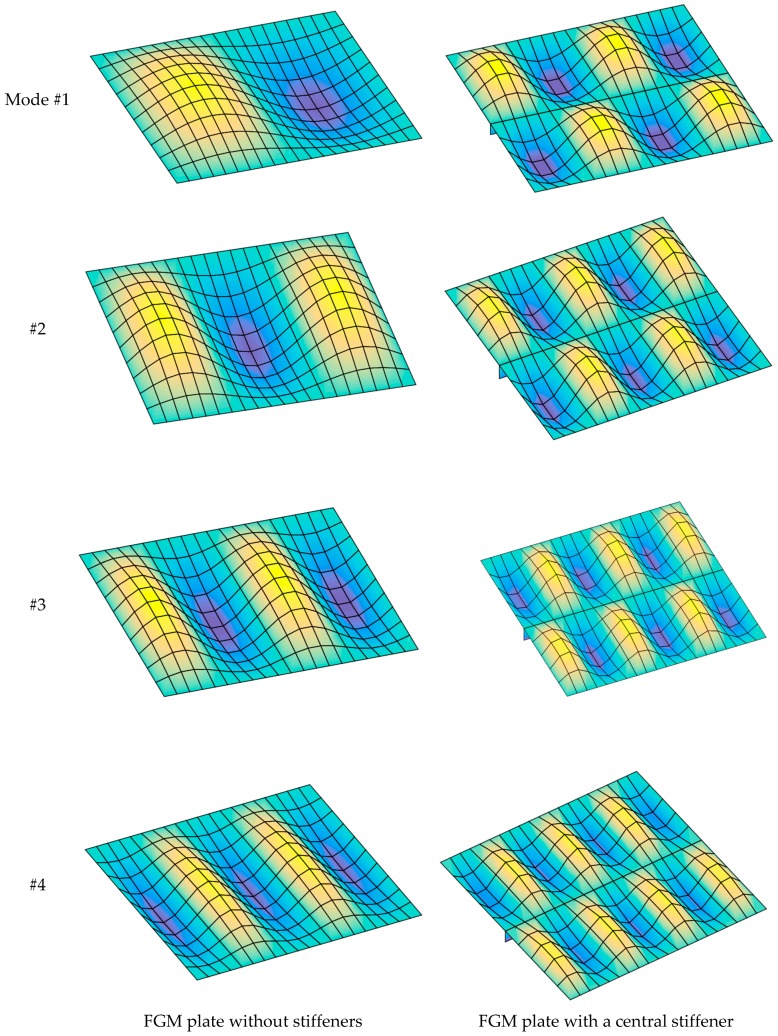
The first four buckling mode shapes of FGM plate with and without a central stiffener (*b*/*h* = 50, *n* = 0.5, Si_3_N_4_/SUS304).

**Figure 8 materials-12-01262-f008:**
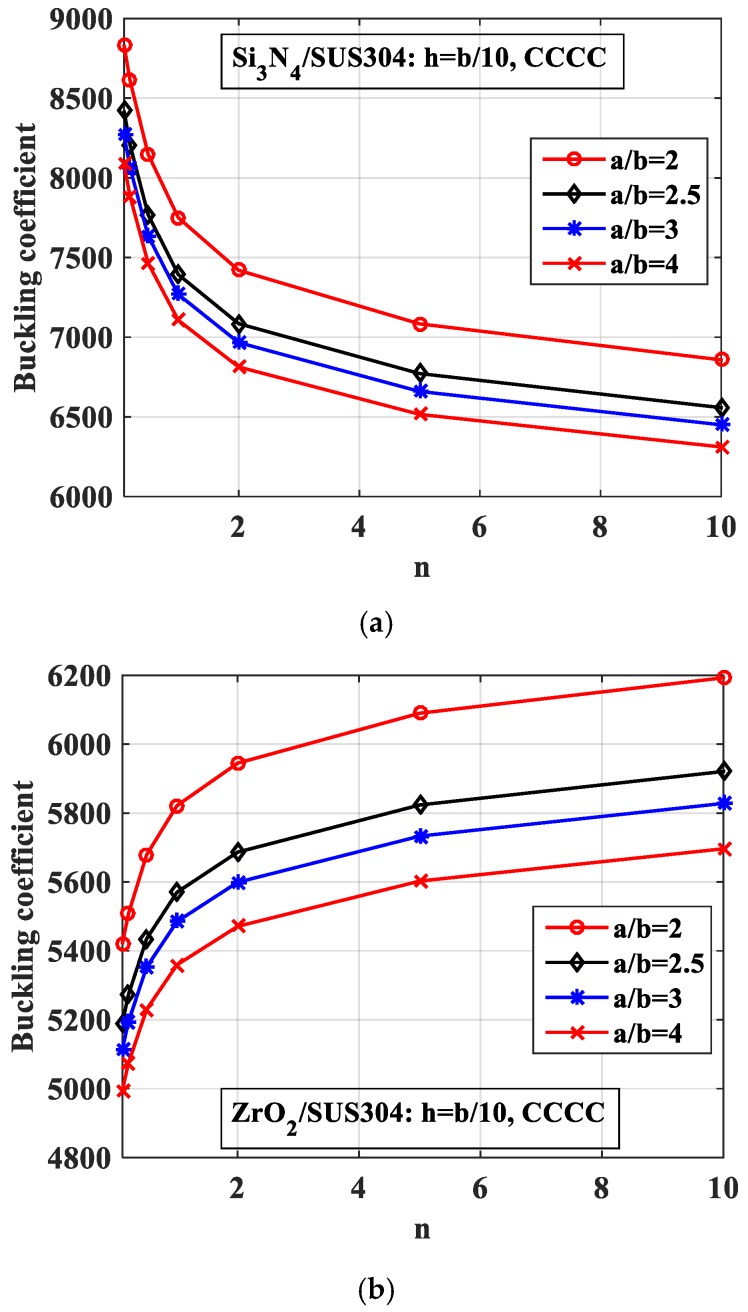
Variation of the buckling coefficient of the plate by the variation of the length-to-width ratio (*a*/*b*). (**a**) Si_3_N_4_/SUS304; (**b**) ZrO_2_/SUS304.

**Figure 9 materials-12-01262-f009:**
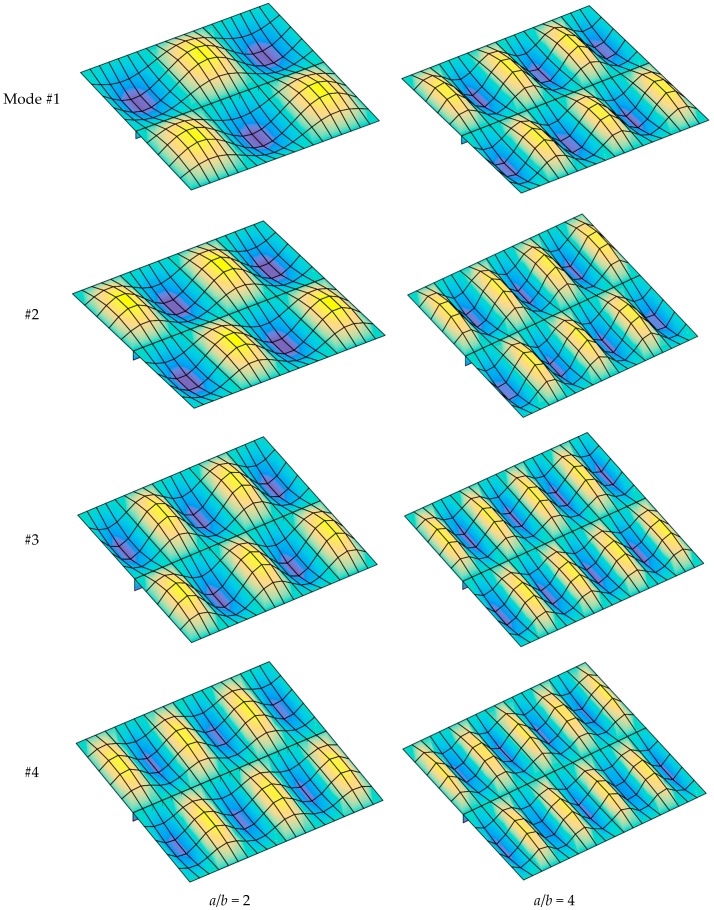
The first four buckling mode shape of stiffened FGM plate (*b*/*h* = 10, *n* = 0.5, Si_3_N_4_/SUS304).

**Figure 10 materials-12-01262-f010:**
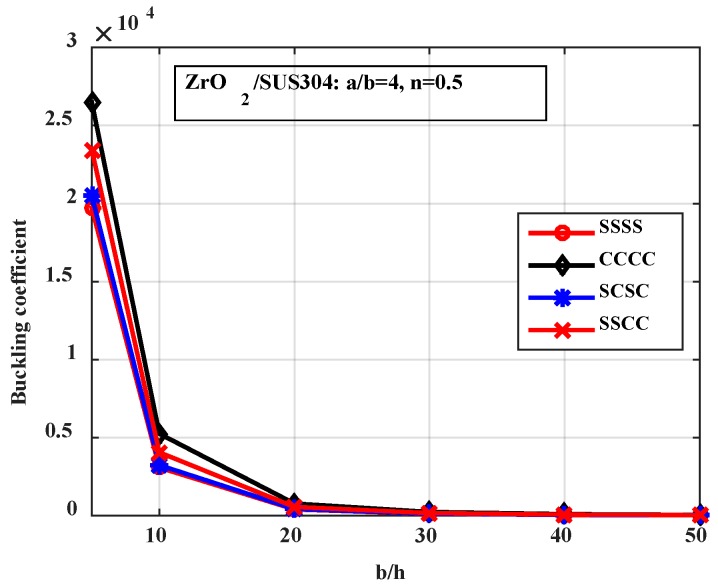
Variation of the buckling coefficient by the boundary condition and the thickness of the plate.

**Figure 11 materials-12-01262-f011:**
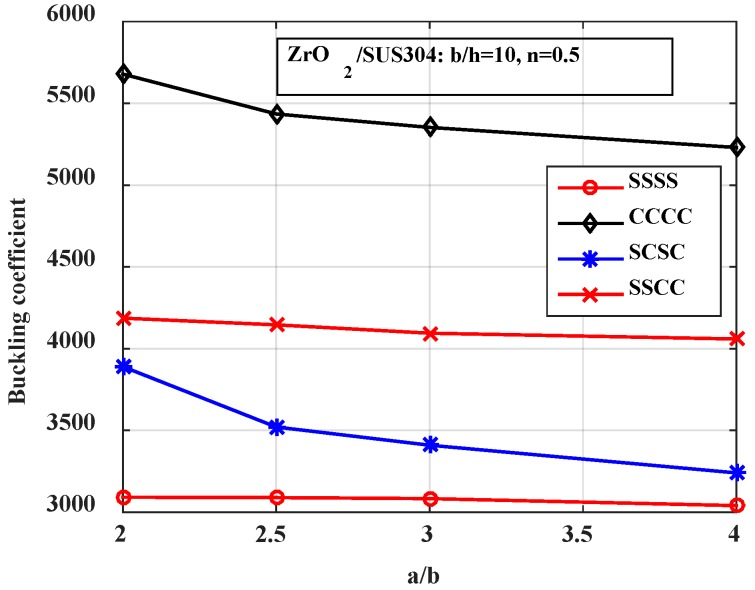
Variation of the buckling coefficient by the boundary condition and the length of the plate.

**Figure 12 materials-12-01262-f012:**
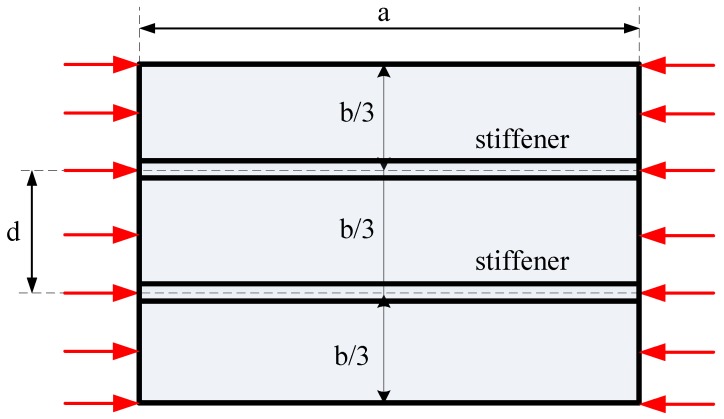
Model of the plate with two parallel stiffeners.

**Figure 13 materials-12-01262-f013:**
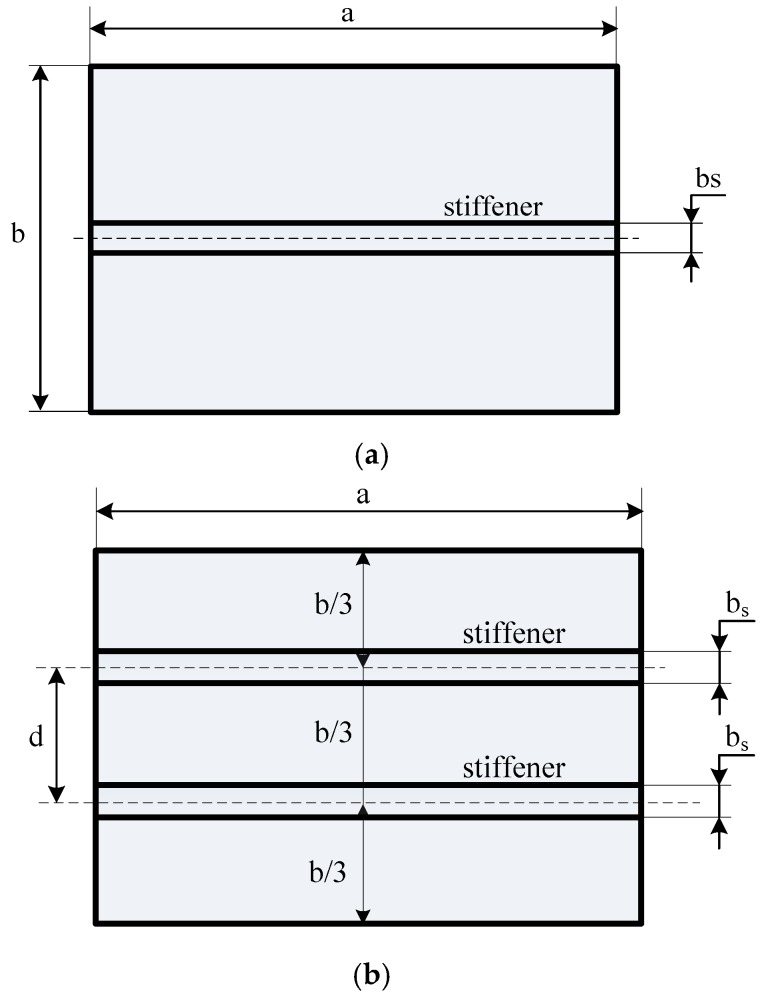
Plate with one stiffener and two stiffeners. (**a**) Plate with one stiffener; (**b**) plate with two stiffeners.

**Figure 14 materials-12-01262-f014:**
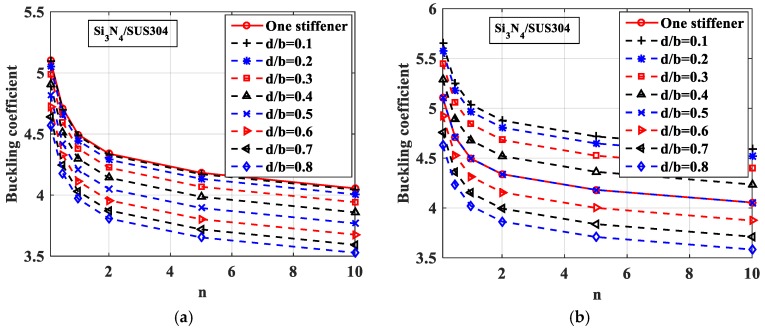
Buckling coefficient of the plate with one stiffener and two stiffeners. (**a**,**c**) Plate with two stiffeners *b_s_* = *b*/200 and plate with one central stiffener; (**b**,**d**) plate with two stiffeners *b_s_* = *b*/100 and plate with one central stiffener.

**Figure 15 materials-12-01262-f015:**
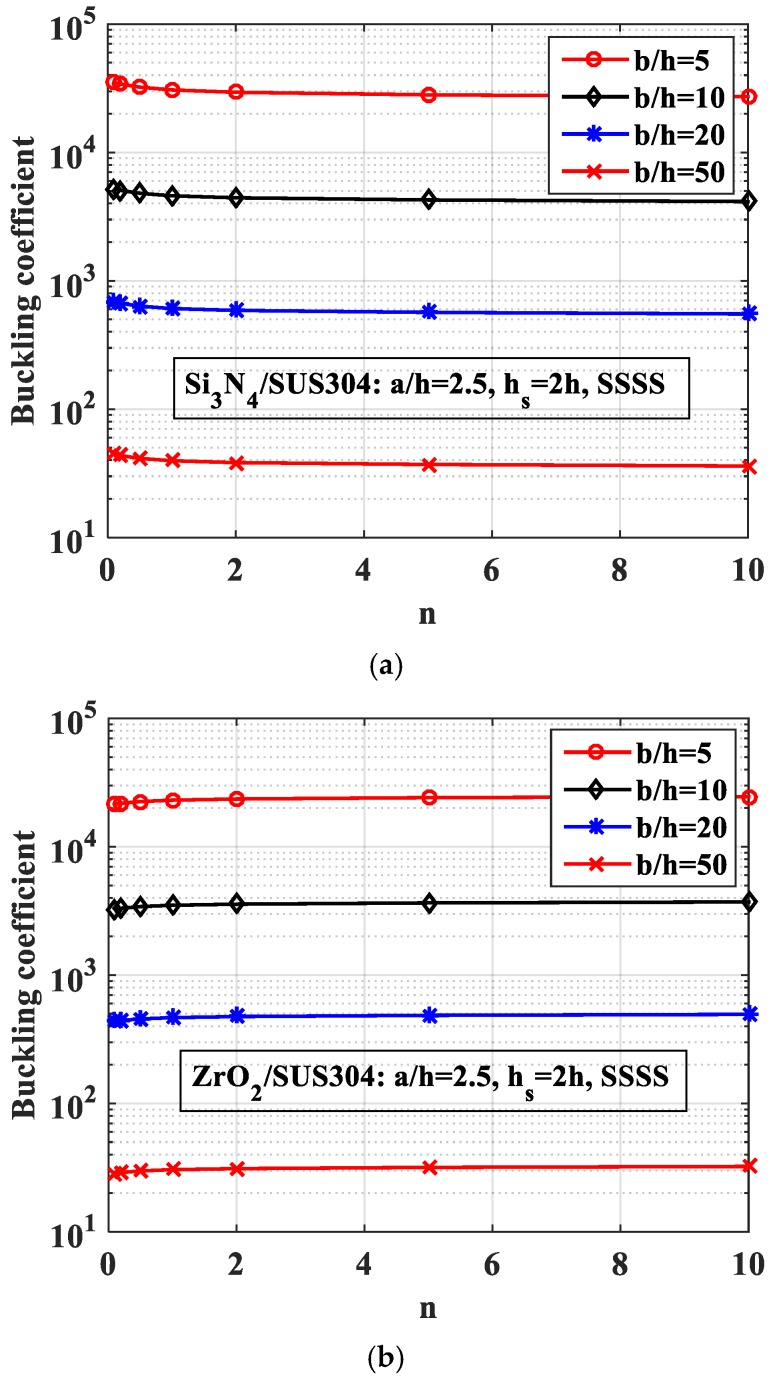
Variation of the buckling coefficient of the plate with two parallel stiffeners by the thickness of the structure. (**a**) Si_3_N_4_/SUS304; (**b**) ZrO_2_/SUS304.

**Figure 16 materials-12-01262-f016:**
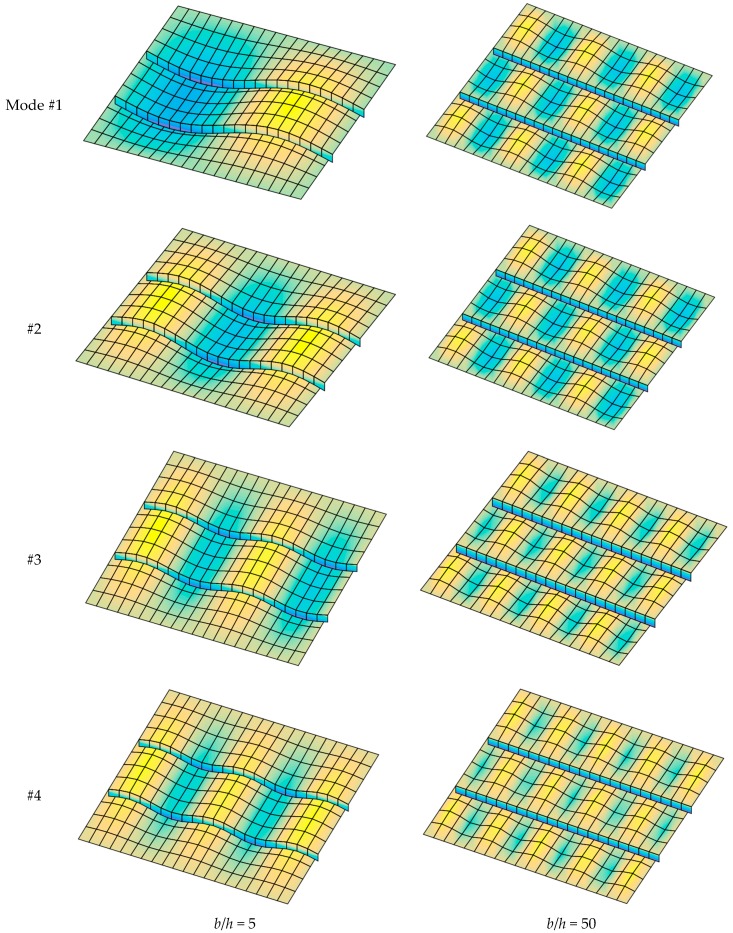
The first four buckling mode shapes of FGM plate with two stiffeners (*n* = 0.5, Si_3_N_4_/SUS304).

**Figure 17 materials-12-01262-f017:**
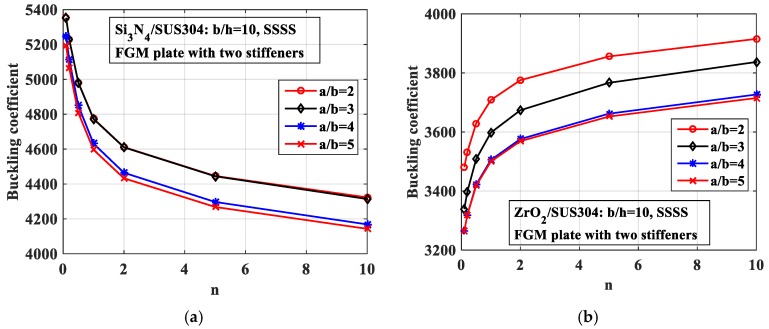
Variation of the buckling coefficient of the plate with two parallel stiffeners by changing of *a*/*b* ratio. (**a**) Si_3_N_4_/SUS304; (**b**) ZrO_2_/SUS304.

**Figure 18 materials-12-01262-f018:**
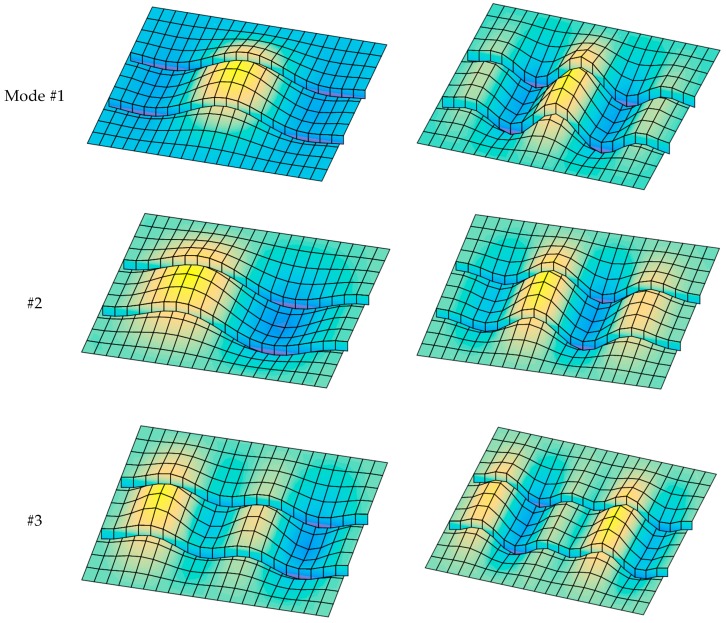
The first four buckling mode shapes of stiffened FGM plate (*b*/*h* = 10, *n* = 0.5, SSSS, Si_3_N_4_/SUS304).

**Table 1 materials-12-01262-t001:** Comparison of the critical buckling load (MN/m) for a simply supported FGM plate (*a* = *b* = 1 m, *h* = *a*/100).

Buckling Load	n = 0	n = 1	n = 2
Meisam [27] (analytical method)	1.3737	0.6847	0.5343
Bodaghi [28] (analytical method)	1.3730	0.6844	0.5340
This work	1.3763	0.6861	0.5353

**Table 2 materials-12-01262-t002:** Buckling coefficient of the clamped plate with a single stiffener.

Buckling Coefficient	Rikards [31]	Rikards [31] Ansys	Mukhopadhyay [29]	Peng [30]	This Work
$k_{buk}$	24.85	23.44	25.46	25.33	26.26

**Table 3 materials-12-01262-t003:** Properties of materials.

Material	Si_3_N_4_	ZrO_2_	SUS304
E (GPa)	322.27	168.06	207.79
ν	0.24	0.298	0.3218

**Table 4 materials-12-01262-t004:** Buckling coefficient of FGM plate with and without a central stiffener (Si_3_N_4_/SUS304, *b*/*h* = 100).

*n*	With Stiffener	*b*/*h* = 100 (Si_3_N_4_/SUS304)							
		*h_s_*/*h*	4	3.5	3	2.5	2	1.5	1
0.1			10.769	9.0043	7.5857	6.4824	5.669	5.1108	4.7654
0.5			9.9746	8.5729	7.1717	6.0785	5.2704	4.7145	4.3698
1			9.5456	8.3323	6.9436	5.8581	5.0543	4.5004	4.1564
2			9.2286	8.1451	6.7696	5.6923	4.8934	4.342	3.9992
5			8.9137	7.9519	6.5918	5.5248	4.7321	4.1843	3.8432
10			8.6595	7.8021	6.4518	5.3912	4.6024	4.0569	3.717

**Table 5 materials-12-01262-t005:** The thickness relationship between Si_3_N_4_/SUS304 plate with and without a central stiffener (*b*/*h* = 100).

*n*	With Stiffener	*b*/*h* = 100 (Si_3_N_4_/SUS304)							
		*h_s_*/*h*	4	3.5	3	2.5	2	1.5	1
0.1	without stiffener	*b*/*h*	74.5	79.0	83.9	89.0	92.5	-	-
0.5			74.1	78.0	83.1	88.0	92.1	-	-
1			73.9	77.4	82.0	87.1	92.1	-	-
2			73.6	76.9	81.6	87.0	91.6	-	-
5			73.5	76.4	80.9	86.3	91.0	-	-
10			73.4	76.1	80.7	85.9	90.1	94.9	-

**Table 6 materials-12-01262-t006:** The mass reduction between Si_3_N_4_/SUS304 plate with and without a central stiffener (*b*/*h* = 100).

*n*	With Stiffener	*b*/*h* = 100 (Si_3_N_4_/SUS304)							
		*h_s_*/*h*	4	3.5	3	2.5	2	1.5	1
0.1	without stiffener		1.0302	1.0126	0.9917	0.9699	0.9636	-	-
0.5			1.1199	1.0954	1.0581	1.0255	1.0029	-	-
1			1.586	1.1329	1.0952	1.0542	1.0172	-	-
2			1.1892	1.1617	1.1176	1.0688	1.0325	-	-
5			1.2110	1.1855	1.1401	1.0872	1.0466	-	-
10			1.2000	1.1963	1.1478	1.0959	1.0595	1.0192	-

**Table 7 materials-12-01262-t007:** Buckling coefficient of FGM plate with and without a central stiffener (Si3N4/SUS304, *b*/*h* = 150).

*n*	with Stiffener	*b*/*h* = 150 (Si_3_N_4_/SUS304)							
		*h_s_*/*h*	4	3.5	3	2.5	2	1.5	1
0.1			2.625	2.2569	1.9623	1.7357	1.5703	1.4577	1.3884
0.5			2.4991	2.135	1.8427	1.6175	1.4527	1.3404	1.2712
1			2.4294	2.068	1.7775	1.5534	1.3892	1.2771	1.208
2			2.3758	2.0173	1.7287	1.5057	1.3422	1.2304	1.1614
5			2.3207	1.9656	1.6794	1.4579	1.2953	1.184	1.1153
10			2.2775	1.9247	1.6399	1.4194	1.2574	1.1465	1.078

**Table 8 materials-12-01262-t008:** The mass reduction between Si_3_N_4_/SUS304 plate with and without a central stiffener (*b*/*h* = 150).

*n*	with Stiffener	*b*/*h* = 150 (Si_3_N_4_/SUS304)							
		*h_s_*/*h*	4	3.5	3	2.5	2	1.5	1
0.1	without stiffener		0.9782	0.9670	0.9501	0.9455	-	-	-
0.5			1.0615	1.0369	1.0163	0.9945	-	-	-
1			1.1077	1.0654	1.0390	1.0127	-	-	-
2			1.1378	1.0976	1.0616	1.0336	-	-	-
5			1.1653	1.1201	1.0789	1.0438	-	-	-
10			1.1816	1.1342	1.0899	1.0488	1.0244	-	-

**Table 9 materials-12-01262-t009:** Buckling coefficient of FGM plate with and without a central stiffener (Si_3_N_4_/SUS304, *b*/*h* = 200).

*n*	With Stiffener	*b*/*h* = 200 (Si_3_N_4_/SUS304)							
		*h_s_*/*h*	4	3.5	3	2.5	2	1.5	1
0.1			0.9807	0.8605	0.7649	0.6919	0.6388	0.6027	0.5807
0.5			0.9292	0.81	0.7149	0.6422	0.5892	0.5533	0.5312
1			0.901	0.7824	0.6878	0.6153	0.5625	0.5266	0.5045
2			0.8796	0.7617	0.6676	0.5954	0.5427	0.5069	0.4849
5			0.8579	0.7409	0.6474	0.5756	0.5231	0.4874	0.4655
10			0.8406	0.7242	0.6311	0.5595	0.5072	0.4716	0.4497

**Table 10 materials-12-01262-t010:** The mass reduction between Si_3_N_4_/SUS304 plate with and without a central stiffener (*b*/*h* = 200).

*n*	With Stiffener	*b*/*h* = 200 (Si_3_N_4_/SUS304)							
		*h_s_*/*h*	4	3.5	3	2.5	2	1.5	1
0.1	without stiffener		0.9358	0.9347	-	-	-	-	-
0.5			1.0259	1.0044	-	-	-	-	-
1			1.0687	1.0306	-	-	-	-	-
2			1.0929	1.0617	1.0322	-	-	-	-
5			1.1194	1.0816	1.0513	-	-	-	-
10			1.1317	1.0960	1.0645	-	-	-	-

**Table 11 materials-12-01262-t011:** Buckling coefficient of FGM plate with and without a central stiffener (ZrO2/SUS304, *b*/*h* = 100).

*n*	with Stiffener	*b*/*h* = 100 (ZrO_2_/SUS304)							
		*h_s_*/*h*	4	3.5	3	2.5	2	1.5	1
0.1			6.7875	6.2948	5.4174	4.4088	3.6519	3.1247	2.7944
0.5			7.1069	6.6127	5.5963	4.5779	3.8152	3.2848	2.9528
1			7.2752	6.7802	5.691	4.6672	3.9013	3.3691	3.0362
2			7.4164	6.9208	5.7701	4.7418	3.9732	3.4396	3.1061
5			7.5918	7.0956	5.8647	4.8323	4.0616	3.5269	3.1932
10			7.7278	7.2314	5.9359	4.9013	4.1296	3.5946	3.2608

**Table 12 materials-12-01262-t012:** The mass reduction between ZrO_2_/SUS304 plate with and without a central stiffener (*b*/*h* = 100).

*n*	with Stiffener	*b*/*h* = 100 (ZrO_2_/SUS304)							
		*h_s_*/*h*	4	3.5	3	2.5	2	1.5	1
0.1	without stiffener		1.1365	1.1406	1.1175	1.0769	1.0410	-	-
0.5			1.1794	1.1792	1.1401	1.0913	1.0567	-	-
1			1.2011	1.1979	1.1519	1.1010	1.0645	-	-
2			1.2178	1.2122	1.1591	1.1069	1.0671	-	-
5			1.2296	1.2223	1.1666	1.1133	1.0647	-	-
10			1.2326	1.2242	1.1650	1.1138	1.0614	-	-

**Table 13 materials-12-01262-t013:** Buckling coefficient of FGM plate with and without a central stiffener (ZrO_2_/SUS304, *b*/*h* = 150).

*n*	with Stiffener	*b*/*h* = 150 (ZrO_2_/SUS304)							
		*h_s_*/*h*	4	3.5	3	2.5	2	1.5	1
0.1			1.8506	1.6201	1.3485	1.1366	0.9798	0.872	0.8051
0.5			1.9447	1.6725	1.3985	1.1851	1.0274	0.9191	0.8520
1			1.9942	1.7001	1.4249	1.2106	1.0525	0.9439	0.8766
2			2.0358	1.7232	1.4469	1.2319	1.0735	0.9647	0.8973
5			2.0876	1.751	1.4737	1.2582	1.0994	0.9905	0.9231
10			2.1152	1.772	1.4941	1.2784	1.1195	1.0106	0.9432

**Table 14 materials-12-01262-t014:** The mass reduction between ZrO_2_/SUS304 plate with and without a central stiffener (*b*/*h* = 150).

*n*	with Stiffener	*b*/*h* = 150 (ZrO_2_/SUS304)							
		*h_s_*/*h*	4	3.5	3	2.5	2	1.5	1
0.1	without stiffener		1.1150	1.1023	1.0640	1.0339	-	-	-
0.5			1.1547	1.1276	1.0871	1.0497	-	-	-
1			1.1755	1.1447	1.1002	1.0589	-	-	-
2			1.1923	1.1506	1.1067	1.0659	-	-	-
5			1.2036	1.1574	1.1119	1.0695	-	-	-
10			1.2074	1.1532	1.1072	1.0683	-	-	-

**Table 15 materials-12-01262-t015:** Buckling coefficient of FGM plate with and without a central stiffener (ZrO_2_/SUS304, *b/h* = 200).

*n*	with Stiffener	*b*/*h* = 200 (ZrO_2_/SUS304)							
		*h_s_*/*h*	4	3.5	3	2.5	2	1.5	1
0.1			0.7114	0.6004	0.5109	0.4417	0.3909	0.3561	0.3347
0.5			0.7335	0.6216	0.5314	0.4618	0.41086	0.376	0.3545
1			0.7452	0.6327	0.5423	0.4725	0.4213	0.3864	0.3649
2			0.755	0.642	0.5513	0.4813	0.4301	0.3951	0.3736
5			0.7667	0.6534	0.5624	0.4923	0.441	0.406	0.3845
10			0.7756	0.6621	0.571	0.5008	0.4495	0.4145	0.3929

**Table 16 materials-12-01262-t016:** The mass reduction between ZrO_2_/SUS304 plate with and without a central stiffener (*b/h* = 200).

*n*	with Stiffener	*b*/*h* = 200 (ZrO_2_/SUS304)							
		*h_s_*/*h*	4	3.5	3	2.5	2	1.5	1
0.1	without stiffener		1.0855	1.0573	1.0300	-	-	-	-
0.5			1.1148	1.0856	-	-	-	-	-
1			1.1299	1.0962	-	-	-	-	-
2			1.1427	1.1068	-	-	-	-	-
5			1.1528	1.1144	-	-	-	-	-
10			1.1507	1.1126	-	-	-	-	-

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
