# Peer review of "Research on the Buckling Behavior of Functionally Graded Plates with Stiffeners Based on the Third-Order Shear Deformation Theory"

_materials, 2019, doi:10.3390/ma12081262_

Reviewer 1 Report

Paper include adaption/implementation of the G.Shi third order plate theory (2007) for modelling buckling behaviour of functionally graded plates with stiffeners. Additionally, the finite element formulation for posed problem is derived and implemented.

The problem considered is complex enough, this theory is not yet widely used.

Authors contribution exists. Suggestion: minor revision.

 Reviewers remarks/suggestions:

1.  The title of the paper should be corrected by omitting “New research on the”.

Paper does not include any new method or theory, also term “new research” is not good itself.

2.  On Page 1:  „this approach is based on a new third-order shear deformation theory (TSDT) without any shear correction factors“

   a)    This theory is introduced  by G.Shi in 2007, thus using term “new” is not so good

   b)     It is well-known that HSDT theories as rule does not need shear correction factors

Thus, more correct text is:

„Current approach is based on a third-order shear deformation theory introduced by G.Shi.“

3.  Through paper, terms “new shear deformation theory” is better to replace with “G.Shi shear deformation theory”

4.  The numbers of formulas from (4)-(8) are in several rows and should be corrected.

5.  The real value of any proposed solution/approach depend how widely it is applicable. In regard to this, please add short discussion or authors vision on applicability/extension of the proposed solution in the case of (can be added in introduction):

   a)       Widely used alternate graduation functions

Exponential-law function

(Chi, S., Chung, Y. International Journal of Solids and Structures, 2006, 43, 3657–3674.)

Mori-Tanaka’s function

(Mori, T., Tanaka, K. Acta Metallurgica, 1973, 21(5), 571–574;

   b)      Nonlinear elastic material models

Lellep, J., Majak, J. Mechanics of Composite Materials,2000, 36 (4), 261−266.

6.       Paper include too much similar figures, which does not provide new information. The number of similar figures should be reduced significantly. Instead some few tables can be added, to provide some more precise numerical data, which can be used in future for comparison of results.

7.       The numerical results are compared by other authors results where different numerical methods are applied. However, in future study it will be interesting to compare results with results obtained by applying other higher order plate theories, but the same numerical methods.

Author Response

Thank you for your comments. Please see attached files

Reviewer 2 Report

The paper aims to study the buckling behavior of stiffened plates made of functionally graded materials in the framework provided by the TSDT. A finite element formulation is proposed for this reason. In the reviewer’s opinion, the topic is quite interesting even if it is not new. Therefore, the authors must emphasize more the novelty of this research. Generally speaking, the paper is not extremely clear and some aspects should be revised. Many readings are required to understand some aspects.

In addition, the following comments and observations should be taken into account:

1. The use of English could be improved throughout the whole manuscript.

2. Some expressions are not well-formatted (see equations 4 and 5).

3. The finite element formulation could be illustrated in a clearer manner. Some comments could be added to explain better the use of both Lagrange and Hermite interpolating functions. Please, clarify the model defined in equations 5-8.

4. Equations 19 introduce the topic of the stiffener. Nevertheless, further details are required since it is not totally clear.

5. The list of reference is inadequate to introduce such a wide topic. The theme of FGM has been widely investigated and many important contributions have been omitted.

6. The authors present many results. It is suggested to focus on the more relevant aspects of the numerical applications.

All things considered, a major revision is suggested.

Author Response

Thank you for your comments. Please see attached files

Round  2

Reviewer 2 Report

As stated in the previous review, the paper deals with an interesting topic but its novelty should be more emphasized, especially in the abstract.

Then, some minor issues concerning the style and the formatting are still included in the manuscript:

1. The numbering on the right of equations (5)-(8) is still not well-formatted. Please, be careful on the parenthesis of the numbering.

2. Equation (14) is cut.

3. The label of figure 2 should be centered.

4. Some typos should be corrected as well (“Hermiteinterpolation functions” should be replaced by “Hermite interpolation functions” and so on…).

Author Response

Thank you for your comment, our justification and explanations are as follows:

1. The numbering on the right of equations (5)-(8) is still not well-formatted. Please, be careful on the parenthesis of the numbering.

Answer: We apologize for these mistakes. The numbers of formulas from (5)-(8) are corrected.

2. Equation (14) is cut.

Answer: Eq. (14) is now ok. 

3. The label of figure 2 should be centered.

Answer: The label of figure 2 is now centered.

4. Some typos should be corrected as well (“Hermiteinterpolation functions” should be replaced by “Hermite interpolation functions” and so on…).

Answer: The typos are corrected.

Last but not least, we would like to express our thankfulness again to the referees for their great comments. All the changes have been highlighted in red color.

Sincerely yours,